

# Living cover crops reduce pesticide residues in agricultural soil

Noé Vandevoorde[1], Igor Turine[1], Alodie Blondel[2], and Yannick Agnan[1]

[1]Earth and Life Institute, UCLouvain, 1348 Louvain-la-Neuve, Belgium
[2]Walloon Agricultural Research Center (CRA-W), 5030 Gembloux, Belgium

**Correspondence:** Noé Vandevoorde (noe.vandevoorde@uclouvain.be)

**Abstract.**

Living cover crops play a key role in reducing nitrogen leaching to groundwater during fallow periods. They also enhance soil microbial activity through root exudates, improving soil structure and increasing organic matter content. While the degradation of pesticides in soil relies primarily on microbial biodegradation, the extent to which cover crops influence this degradation remains poorly quantified. In this paper we (1) monitored pesticide residue levels in soil and soil solution under two different cover crop densities and (2) correlated the observed reductions with physicochemical properties of the active substances. Our results show that thin cover crops ($0.4\,\mathrm{t_{DM}\,ha^{-1}}$) reduce pesticide leaching 80 days after sowing compared to bare soil, retaining the residues in the microbiologically active topsoil. In addition, well-developed cover crops ($1\,\mathrm{t_{DM}\,ha^{-1}}$) reduce soil pesticide contents by more than $33\,\%$ for compounds with low to high water solubility ($s \leqslant 1400\,\mathrm{mg\,L^{-1}}$) and low to moderate soil mobility ($\mathrm{K_{oc}} \geqslant 160\,\mathrm{mL\,g^{-1}}$). This effect is probably due to enhanced pesticide degradation of the retained pesticide in the rhizosphere. These results confirm previous studies on individual compounds, individual cover crop types and individual soil compartments, while providing new thresholds for physicochemical properties associated with significant pesticide degradation. By directly enhancing pesticide degradation within the soil compartment where pesticides are applied, cover crops limit their transfer to other environmental compartments, particularly groundwater.

## 1 Introduction

Pesticides play a major role in modern agriculture, helping to stabilise crop yields, optimise farm labour and income, and ensure overall agricultural production (Cooper and Dobson, 2007; Oerke, 2006). However, their use is associated with multiple —and well-documented— negative impacts on the environment and human health (Damalas and Koutroubas, 2016; Kim et al., 2017; Mandal et al., 2020; Stoate et al., 2001). Among these, the widespread contamination of ecosystems and consequent degradation of ecosystem services (Leenhardt et al., 2023; Power, 2010; Silva et al., 2019) directly affects the quality of drinking water supplies (Joerss et al., 2024; Pedersen et al., 2016; Syafrudin et al., 2021), poses risks to general human health (Gerken et al., 2024; Rani et al., 2021; Scorza et al., 2023; Shekhar et al., 2024) and results in significant social costs (Alliot et al., 2022; Bourguet and Guillemaud, 2016).

Pesticides applied to plants and agricultural soils undergo various environmental fates: (1) they may be degraded by photolysis, hydrolysis, abiotic oxidation or biodegradation into a range of degradation products; (2) they may be bound to soil minerals and organic matter or be absorbed by plant roots; or (3) they may be transferred off-site by volatilisation, run-off,





erosion or leaching to groundwater bodies. While these processes reduce pesticide content in agricultural soil, they contribute to diffuse contamination of other environmental compartments. This issue is further exacerbated by the persistence of pesticide residues in soil long after application, sustaining diffuse contamination even after the pesticides have been banned. For exam-

ple, atrazine residues continue to leach into groundwater decades after it was banned in 2004 (de Albuquerque et al., 2020). Similarly, despite being banned in 1993, chlordecone adsorbed on soil particles is currently being transported to surface and groundwater bodies by soil erosion (enhanced by bare soils resulting from contemporary glyphosate applications) and desorption (facilitated by site competition with glyphosate due to its chelating properties; Sabatier et al., 2021; Bemelmans et al., 2023). These challenges underline the need to explore strategies to limit the persistence and mobility of pesticides in topsoil as

soon as possible after application. Among these strategies, bio- and phyto-remediations offer a promising avenue.

Bioremediation transforms contaminants into non-toxic substances through the activity of soil microorganisms. Phytoremediation extends this process, encompassing plants and their rhizosphere (Cycoń et al., 2017; Eevers et al., 2017; Jia et al., 2023). This involves (1) rhizodegradation, rhizostabilisation and rhizofiltration which degrade, stabilise or concentrate contaminants near the roots, respectively, and (2) plant uptake and metabolism, aided by endophytic microorganisms. In particular,

rhizofiltration is induced by soil water flux driven by the plant evapotranspiration (Tarla et al., 2020). Root exudates provide nutrients that stimulate microbial activity and promote synergistic interactions within rhizospheric microbial communities, enhancing the degradation of persistent compounds. In addition, plant and microbial enzymes co-degrade pesticides in the rhizosphere, with root dynamics improving soil aeration and facilitating oxidative degradation (Eevers et al., 2017; Jia et al., 2023; McGuinness and Dowling, 2009). Rhizoremediation can thus be considered as a biostimulation strategy in which plants

stimulate native microbial communities via root exudates, amplifying pesticide bioremediation (Cycoń et al., 2017; Tarla et al., 2020). Phytoremediation approaches are particularly suited to mitigating diffuse pollution from cumulative agricultural applications, offering scalable, cost-effective solutions that stabilise and degrade pesticides while preventing their transfer to other environmental compartments (Eevers et al., 2017; McGuinness and Dowling, 2009; Tarla et al., 2020).

Originally introduced to reduce soil erosion and nitrate leaching (as catch crops), cover crops are closely related to the

principles of phytoremediation. By maintaining a living plant cover during the fallow period, they stimulate soil microbial activity and offer a practical way to integrate phytoremediation into annual agricultural cycles without taking land out of production. In addition to their biostimulative effects, cover crops induce physical, chemical and biological changes in the soil environment and contribute to ecosystem services such as nutrient cycling, water regulation or pest and disease suppression (Dabney et al., 2001; Hao et al., 2023; Justes and Richard, 2017; Reeves, 1994). These changes also influence pesticide

dynamics, including mobility, retention and degradation within the soil. While the effects in situ of established cover crops on newly applied pesticides have been widely studied (e.g. Cassigneul et al., 2015, 2016; Perkins et al., 2021; Whalen et al., 2020), research on the effects of newly sown cover crops on existing pesticide residues remains limited.

In this limited research, studies suggest several mechanisms by which cover crops can reduce pesticide transport, including increasing soil organic matter, enhancing microbial activity and improving soil structure. These processes contribute to greater

pesticide adsorption, faster degradation and reduced leaching. For example, a one year field study by Bottomley et al. (1999) showed that winter rye (*Secale cereale*) enhanced subsurface microbial activity, thereby promoting the mineralisation of 2,4-D.



Similarly, multi year field studies by Potter et al. (2007) and White et al. (2009) reported significant reductions in groundwater concentrations and soil contents, respectively, under sunn hemp (*Crotalaria juncea*) cover crops compared to bare soil, with reductions of up to 33 and $41\,\%$ for atrazine and metolachlor, respectively. However, these studies focused on individual

molecules, specific cover types and single soil compartment (soil or soil solution), limiting the generalisability of their results.

Long-term field experiments, such as those conducted by Alletto et al. (2012) and Pelletier and Agnan (2019), have extended these studies by examining multiple factors influencing pesticide retention and mobility, in both soil and soil solution. Alletto et al.'s study, conducted over four years, showed that cover crops such as oats (*Avena sativa*) could reduce isoxaflutole losses by 25 to $50\,\%$ compared to bare soil. They highlighted the importance of soil organic carbon and cover biomass production

in reducing leaching, with cover crops producing over $2\,\mathrm{t_{DM}\,ha^{-1}}$ significantly reducing leaching in contrast to no effects observed at $0.3\,\mathrm{t_{DM}\,ha^{-1}}$ (DM: dry matter). These results illustrate the potential of cover crops to improve soil properties, increasing the travel time of pesticides through biologically active soil layers and facilitating their degradation before reaching groundwater. Pelletier and Agnan extended this research to 32 active substances and soil solution analyses. They identified organic carbon content and evapotranspiration from cover crops as critical factors in the retention of pesticides in the biologi-

cally active layers. In addition, they observed a resurgence of certain molecules under fully developed cover crops, suggesting that evapotranspiration can bring back up substances that have started to leach down in the soil profile. This underlines the criticality of the transition between (cash) crop and cover crop periods, when reduced evapotranspiration can lead to increased leaching before the cover crop has had time to take full effect. Although five different cover crop mixes were grown, data in Pelletier and Agnan's study were insufficient to make comprehensive comparison between them.

Despite progress in the literature, several limitations remain: (1) bioremediation studies often prioritise reactor processes for highly contaminated soils rather than field applications for diffuse pollution; (2) few studies examine the combined effects of cover crops and native microbial communities without inoculation; and (3) field research is limited to a narrow range of pesticide molecules and cover crops, with inconsistent assessments of soil compartments. These gaps prevent a broader understanding of the general applicability of cover crop remediation strategies for different pesticide molecules.

To address these gaps, we conducted a controlled, three months greenhouse experiment. Our aim was to evaluate the ability of newly sown cover crops to influence the dynamics of pesticide residues in soil and soil solution under conditions that bridge the gap between reactor-based studies and field applications, without microbial inoculation. Specifically, we monitored the evolution of 18 active substances over time under three modalities: two contrasting densities of living cover crops and a control (bare soil), thereby extending the scope of previous field studies. In addition, we aimed to correlate the observed

evolution of pesticide levels with the physicochemical properties of the molecules. Our main hypothesis was that cover crops reduce pesticide leaching by altering soil water fluxes through evapotranspiration, concentrating pesticides near the roots and prolonging their retention in the microbiologically active rhizosphere where they can be metabolised more rapidly. Following Tarla et al. (2020), we hypothesised that these mechanisms are more important than plant uptake and have therefore focused on soil and soil solution, excluding plant tissue analysis from the study. This approach aimed to provide preliminary evidence

for the categorisation of pesticides affected by cover crops and to refine hypotheses on evapotranspiration and microbiological degradation.





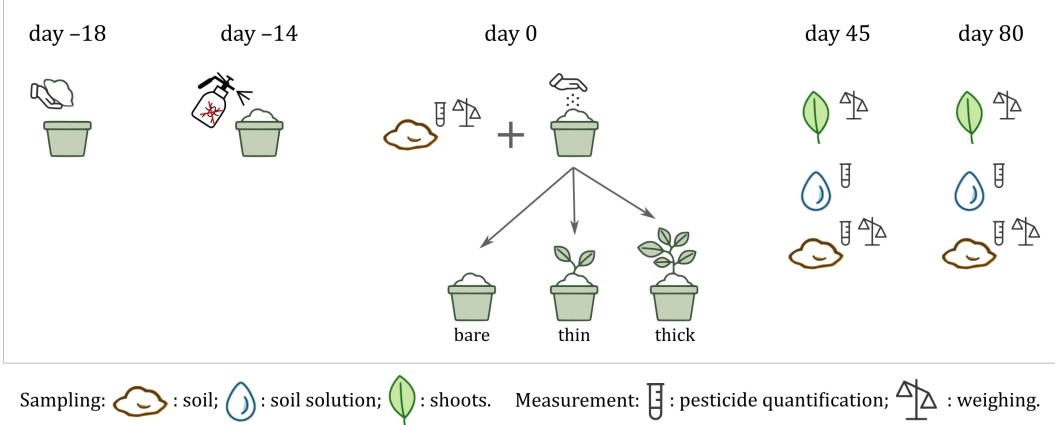

**Figure 1.** Experiment setup, sampling and measurement timeline; day 0 corresponds to 5 January 2024.

## 2 Materials and Methods

In this paper, we present our numerical results with their standard deviation and propagated uncertainties as: value $\pm_{\mathrm{sd}}$ standard deviation $\pm_{\Delta}$ (propagated) measurement uncertainty. When calculating a value $f(x_1, \ldots, x_n)$ from experimental data $x_i$, the propagation of uncertainties $\Delta f$ due to random and independent measurement errors $\Delta x_i$, is determined using the general propagation formula:

$$\Delta f(x_1, \ldots, x_n) = \sqrt{\sum_{i=1}^{n} \left( \frac{\partial f}{\partial x_i} \Delta x_i \right)^2} \tag{1}$$

### 2.1 Experimental setup

The soil was collected from the top $30\,\mathrm{cm}$ of an agricultural plot following a white mustard seed crop (UCLouvain University Farms, Corroy-le-Grand, Belgium; $50.6740°$ N, $4.6368°$ E) on 18 December 2023 (day $-18$; Fig. 1). It constituted a silty soil developed on Quaternary loess characterised by slightly acidic conditions ($\mathrm{pH_{H_2O}} = 6.1$), low total carbon content ($0.89\,\%$), balanced carbon/nitrogen ratio ($\mathrm{C/N} = 9$) and a CEC of $11.1\,\mathrm{cmol_c\,kg^{-1}}$. To avoid pesticide contamination, the soil was taken from a certified organic plot. Plants and debris were manually removed from the collected soil, which was then mixed and placed in $10\,\mathrm{L}$ plastic pots ($0.07\,\mathrm{m}^2$ area), each containing $9.64 \pm_{\mathrm{sd}} 0.40 \pm_{\Delta} 0.02\,\mathrm{kg}$ of fresh soil ($n = 35$). The pots were then transferred to the greenhouse.

To simulate pesticide residues from a previous crop, a mixture of 13 formulated pesticide products (containing 18 different known ingredients: 11 herbicides, 5 fungicides, 1 insecticide and 1 safener) was sprayed on the pot's bare soil in the greenhouse on 22 December 2023 (day $-14$) at the maximum authorised dose (across all authorised crops; Table 1; for details, see Table S1 in the Supplement). The composition of the formulated products and the maximum doses authorized were obtained from *phytoweb.be*, the official website of the Belgian Federal Public Services for Health, Food Chain Safety and the Environment for



**Table 1.** 18 applied active substances (day $-14$), with corresponding applied quantities (q).

| active substance | type | formulated product(s) | quantity (q, in $\mathrm{\mu g\,kg^{-1}_{fresh\,soil}}$) | | |
|---|---|---|---|---|---|
| clopyralid | h | *Bofix* | 58 | $\pm_\Delta$ | 3 |
| cloquintocet-mexyl | h | *Axial*, *Capri*, *Frimax* | 30 | $\pm_\Delta$ | 1 |
| fenpicoxamid | f | *Aquino* | 73 | $\pm_\Delta$ | 3 |
| flonicamid | i | *Afinto* | 116 | $\pm_\Delta$ | 5 |
| florasulam | h | *Primus* | 3.6 | $\pm_\Delta$ | 0.2 |
| fluroxypyr | h | *Bofix*, *Frimax* | 213 | $\pm_\Delta$ | 9 |
| fluxapyroxad | f | *Mizona*, *Revytrex* | 189 | $\pm_\Delta$ | 8 |
| halauxifen-methyl | h | *Frimax* | 4.5 | $\pm_\Delta$ | 0.2 |
| iodosulfuron-methyl-sodium | h | *Mesiofis Pro* | 2.18 | $\pm_\Delta$ | 0.09 |
| MCPA | h | *Bofix* | 580 | $\pm_\Delta$ | 30 |
| MCPB | h | *Butizyl* | 1450 | $\pm_\Delta$ | 60 |
| mefenpyr-diethyl | s | *Mesiofis Pro* | 33 | $\pm_\Delta$ | 1 |
| mefentrifluconazole | f | *Revytrex* | 145 | $\pm_\Delta$ | 6 |
| mesosulfuron-methyl | h | *Mesiofis Pro* | 10.9 | $\pm_\Delta$ | 0.5 |
| pinoxaden | h | *Axial* | 44 | $\pm_\Delta$ | 2 |
| pyraclostrobin | f | *Comet New*, *Mizona* | 650 | $\pm_\Delta$ | 30 |
| pyroxsulam | h | *Capri* | 14.2 | $\pm_\Delta$ | 0.6 |
| tebuconazole | f | *Tebusip* | 550 | $\pm_\Delta$ | 20 |

h: herbicide; f: fungicide; i: insecticide; s: safener.

plant protection and fertilising products. The formulated pesticides were selected on the basis of the contrasted physicochemical properties of the active substances, their availability at the University Farms, their possible quantification using a single multi-residue analysis and excluding any root herbicides that could inhibit the germination and growth of the experimental cover crops.

120    Three cover modalities were tested (Fig. 1). Two types of cover crops with rapid growth: (1) 10 pots with a *thick* winter spelt (*Triticum spelta*) cash crop and (2) 10 pots with a *thin* catch crop multi-species cover mix (20 % buckwheat, *Fagopyrum esculentum*; 20 % phacelia, *Phacelia tanacetifolia*; 20 % vetch, *Vicia villosa*; and 40 % white mustard, *Sinapis alba*; seed w/w); in addition to 15 pots kept bare as a control (for a total of 35 pots in the experiment). In the following, we refer to the thick and thin cover crops as *cover types*, while cover types together with the control are collectively referred to as *cover modalities*. The

125    thick and thin cover types were sown on 5 January 2024 (day 0) at a density of $191 \pm_\mathrm{sd} 12 \pm_\Delta 1\,\mathrm{kg_{seeds}\,ha^{-1}}$ ($n = 10$) and $147 \pm_\mathrm{sd} 3 \pm_\Delta 1\,\mathrm{kg_{seeds}\,ha^{-1}}$ ($n = 10$), respectively. They reached a shoot biomass of $0.43 \pm_\mathrm{sd} 0.04 \pm_\Delta 0.07\,\mathrm{t_{DM}\,ha^{-1}}$ and $0.25 \pm_\mathrm{sd} 0.08 \pm_\Delta 0.04\,\mathrm{t_{DM}\,ha^{-1}}$, respectively, on day 45 ($n = 5$), and a shoot biomass of $1.12 \pm_\mathrm{sd} 0.02 \pm_\Delta 0.18\,\mathrm{t_{DM}\,ha^{-1}}$ and $0.36 \pm_\mathrm{sd} 0.09 \pm_\Delta 0.06\,\mathrm{t_{DM}\,ha^{-1}}$, respectively, on day 80 ($n = 5$).





The pots were kept in a greenhouse maintained at $20.8 \pm_{\mathrm{sd}} 1.6\,^{\circ}\mathrm{C}$ and $55 \pm_{\mathrm{sd}} 11\,\%$ humidity, with 12 hours of light per day.
They were watered with rain water twice a week at an average rate of ca. $1\,\mathrm{L}$ per week, corresponding to an average rainfall of $14\,\mathrm{mm}\,\mathrm{week}^{-1}$, leading to an average soil moisture of $79.16 \pm_{\mathrm{sd}} 1.10 \pm_{\Delta} 0.01\,\%_{\mathrm{DM}}$ (w/w; $n = 35$).

Raw data regarding the experimental setup are detailed in Table S2 in the Supplement.

## 2.2 Soil, soil solution and plant sampling

An initial soil sampling was performed on five control pots at the time of sowing (day 0; Fig. 1). Subsequently, the sampling was carried out in five pots per cover modality on 19 February 2024 (day 45) and on 25 March 2024 (day 80). On days 45 and 80, three types of samples were collected per pot: (1) plant shoots (for biomass quantification), (2) soil solution sample (for pesticide quantification) and (3) soil sample (for pesticide quantification).

Plant shoots were sampled by cutting the cover at the soil surface. After removal of any dirt, the plant parts were dried in an oven at $60\,^{\circ}\mathrm{C}$ for $24\,\mathrm{h}$, then weighed.

Soil solution was sampled using rhizons (micro suction cups consisting of a $2.5\,\mathrm{mm}$ diameter, $10\,\mathrm{cm}$ long hydrophilic polyether sulphone membrane with a $0.15\,\mu\mathrm{m}$ porosity; 19.21.21F, *Rhizosphere®*, Wageningen, Netherlands), installed vertically in the top $10\,\mathrm{cm}$ in the centre of each pot. Soil solution samples were collected using $60\,\mathrm{mL}$ polypropylene syringes (BD Plastipak luer lock) manually activated to create a suction of ca. $-700\,\mathrm{hPa}$ maintained for $16\,\mathrm{h}$ using a wooden wedge, $8\,\mathrm{h}$ after a $1\,\mathrm{L}$ watering. Five replicates were collected per modality for each sampling, except for the control on day 45 and the thin cover on day 80 where only four replicates were collected due to faulty rhizons, connecting pipes and/or syringes ($6.7 \pm_{\mathrm{sd}} 5.8\,\%$ drop-out rate per cover modality). Samples were then transferred to glass vials and kept in the dark in a cold storage ($4\,^{\circ}\mathrm{C}$) until analysis.

When multiple sample types were collected (day 45 and day 80), soil was sampled last, after the plant shoots and soil solution. Each pot was individually emptied into a large container to remove the main roots and to thoroughly mix the soil. From this, $1\,\mathrm{kg}$ of fresh soil was sampled on day 0 and day 45, and $200\,\mathrm{g}$ on day 80. Fresh soil samples were then frozen at $-18\,^{\circ}\mathrm{C}$ and kept in the dark until analysis. An extra $500\,\mathrm{g}$ fresh soil sample was collected from each pot to assess soil moisture content (MC) by weighing the soil mass before and after drying in an oven at $70\,^{\circ}\mathrm{C}$ for $48\,\mathrm{h}$: $\mathrm{MC} = (\mathrm{m}_{\mathrm{fresh\ soil}} - \mathrm{m}_{\mathrm{dried\ soil}})/\mathrm{m}_{\mathrm{fresh\ soil}}$.

## 2.3 Pesticide quantification

Soil and soil solution samples were analysed at the laboratory of the Walloon Agricultural Research Centre (CRA-W) in Gembloux, Belgium, for quantification of the 18 applied active substances. No metabolites were quantified. Frozen soil samples were thawed, extracted by QuEChERS and analysed by liquid chromatography (LC) coupled to a quadrupole time-of-flight mass spectrometer (QTOFMS). Soil solution samples were analysed within seven days after collection, extracted with acetonitrile, filtered and analysed on the same LC-QTOFMS instrument. Detail of the analytical method is given in the Supplement S2.

Raw quantification data and limits of quantifications (LQ) are available in Tables S3 and S4 in the Supplement. For data analysis, concentrations below the LQ (<LQ) were assigned a value of $2\,\mathrm{LQ}/3$ and non-detected (ND) values were assigned $\mathrm{LQ}/3$. Throughout the paper, quantifications of active substance in soil samples are expressed as active substance mass per



unit fresh soil mass ($\mu g_{\text{active substance}}\,kg^{-1}_{\text{fresh soil}}$), while in soil solution samples they are expressed as active substance mass per unit soil solution volume ($\mu g_{\text{active substance}}\,L^{-1}_{\text{soil solution}}$).

Residual moisture retained in micropores after gravitational drainage results in fresh soil samples containing both active
substances adsorbed to soil particles and those dissolved in the residual soil solution. For low solubility compounds, this residual solution has minimal effect on quantification. However, for highly soluble, low-volatility substances (e.g. flonicamid, pyroxsulam), their concentration in the residual solution may exceed their adsorbed content and thus bias the analysis. Drying soil samples prior to analysis does not solve this problem, as low-volatility compounds remain and the drying process may volatilise other substances, further biasing the results. This limitation applies to any study quantifying pesticides in soil samples
and affects any comparison with soil solution samples. In this study, this bias prevented the determination of a total mass balance of the active substances by simply combining the content from the soil samples with the concentration from the soil solution samples, as the residual soil solution would effectively be double counted. Nevertheless, in order to allow a direct comparison of the levels of active substances between the two compartments, we have converted the concentrations in soil solution to equivalent fresh soil content (in $\mu g\,kg^{-1}$) by multiplying them by the fraction of soil solution per unit mass of fresh
soil, bearing in mind that the soil content also includes some of the soil solution concentration.

## 2.4  Pesticide properties data source and data treatment

Physicochemical properties of the active substances and threshold interpretations were extracted from the Pesticide Properties DataBase (PPDB; Lewis et al., 2016) on 3 May 2024 and are summarised in Tables S5 and S6 in the Supplement. These properties include: typical soil persistence (DT50$_{\text{soil}}$, in days) and soil sorption coefficient (K$_{\text{oc}}$, in $mL\,g^{-1}$) for the persistence
and mobility in soil, respectively; water solubility at $20\,^{\circ}C$ (s, in $mg\,L^{-1}$) and groundwater ubiquity score (GUS, dimensionless) for the transfer to soil solution and tendency to leach; vapour pressure at $20\,^{\circ}C$ (p, in $mPa$) and Henry's law constant (k$_{\text{H}}$, in $Pa\,m^{3}\,mol^{-1}$) for the transfer to air; n-octanol–water partition coefficient (i.e. lipophilicity) at $pH\,7$ and $20\,^{\circ}C$ (K$_{\text{ow}}$, dimensionless), bioconcentration factor (BCF, in $L\,kg^{-1}$) and relative molecular mass (m, dimensionless) for the uptake into plants.

Data pre-analyses were performed in MS Excel. Further data analyses and visualisations were performed in RStudio (R 4.4.2, R Core Team, 2024; RStudio 2024.09.1).

Interquartile range outlier analysis conducted per sampling date and compartment (across all modalities) showed that a minority (no more than five) of the 18 active substances were affected by outlier values per sample. Consequently all samples were retained in the dataset and no outlier were excluded.

Principal component analysis (PCA) was performed to assess patterns in the quantification data across compartments, modalities and sampling dates. Prior to analysis, the data were subjected to a centred log-ratio transformation using the *R* function `compositions::clr` (van den Boogaart et al., 2005) to account for compositional constraints. The PCA was then performed using `FactoMineR` (Lê et al., 2008), ensuring that data were centred and scaled, and the results were visualised using `factoextra` (Kassambara and Mundt, 2016) and `ggplot2` (Wickham, 2016).



Standard deviation for the differences in active substance content between cover modalities (cover types versus control) was calculated as the propagation of the standard deviations of the cover type and the control (with no correlation factor as the cover modality samples were unpaired):

$$\sigma_{\text{difference}} = \sqrt{\sigma_{\text{type}}^2 + \sigma_{\text{control}}^2} \qquad (2)$$

To assess whether the active substance content differences were statistically significant, individual unilateral t-tests were
performed. These tests evaluated whether the concentration difference between the cover type and the control was different (positive or negative) than zero.

## 3    Results and Discussion

### 3.1    Active substance behaviour by compartment

#### 3.1.1    Soil content

Application rates (day $-14$) ranged from $2.18 \pm_\Delta 0.09\,\mu g\,kg^{-1}$ (iodosulfuron-methyl-sodium) to $1450 \pm_\Delta 60\,\mu g\,kg^{-1}$ (MCPB). By day 0, average active substance contents in soil samples (in all modalities) ranged from $0.25 \pm_{\text{sd}} 0.20\,\mu g\,kg^{-1}$ (pinoxaden) to $730 \pm_{\text{sd}} 260\,\mu g\,kg^{-1}$ (MCPA), corresponding to residues from $0\,\%$ (no detection: iodosulfuron-methyl-sodium and mefenpyr-diethyl) to $130 \pm_\Delta 50\,\%$ (MCPA) of the initial applied mass, with a median of $48\,\%$ over the 18 active substances. All but three substances (pinoxaden, iodosulfuron-methyl-sodium and mefenpyr-diethyl) were quantified in all samples. In particu-
lar, seven active substances (clopyralid, fluroxypyr, fluxapyroxad, MCPA, mefentrifluconazole, mesosulfuron-methyl and tebuconazole) showed residue levels compatible with $100\,\%$ of the initial mass, linked to high application rate ($q \geqslant 145\,\mu g\,kg^{-1}$), very low volatility ($p < 5\,10^{-5}\,\text{mPa}$) and/or moderate to long persistence in soil ($\text{DT50}_{\text{soil}} > 30\,\text{d}$). In contrast, pinoxaden, iodosulfuron-methyl-sodium and mefenpyr-diethyl, characterised by low application rate ($q < 5\,\mu g\,kg^{-1}$), high water solubility ($s > 1000\,\text{mg}\,\text{L}^{-1}$) and/or short soil persistence ($\text{DT50}_{\text{soil}} < 30\,\text{d}$), had quantification rates of 80, 20 and $0\,\%$, respectively.
By day 45, soil contents had decreased from below $0.20\,\mu g\,kg^{-1}$ (cloquintocet-mexyl and pinoxaden; lowest LQ) to $310 \pm_{\text{sd}} 80\,\mu g\,kg^{-1}$ (tebuconazole). This corresponded to residues from $0\,\%$ (no detection) to $62 \pm_\Delta 15\,\%$ (fluxapyroxad) of the initial applied mass, with a median below $0.5\,\%$. This aligns with literature showing that most pesticide loss occurs within the first weeks after application via evaporation, photolysis and hydrolysis (Bedos et al., 2002; Ferrari et al., 2003; Gish et al., 2011). Seven active substances (fenpicoxamid, fluxapyroxad, MCPA, mefentrifluconazole, mesosulfuron-methyl, pyraclostrobin and
tebuconazole) were quantified in all samples, exhibiting at least two of the following characteristics: high application rates ($q \geqslant 145\,\mu g\,kg^{-1}$), low water solubility ($s < 10\,\text{mg}\,\text{L}^{-1}$), high soil sorption ($K_{\text{oc}} > 4000\,\text{mL}\,\text{g}^{-1}$) and/or long soil persistence ($\text{DT50}_{\text{soil}} > 100\,\text{d}$) —except for MCPA, which has a high solubility ($s = 250000\,\text{mg}\,\text{L}^{-1}$) but was applied at the third highest rate ($q = 580\,\mu g\,kg^{-1}$), leaving detectable residues. Five active substances (clopyralid, flonicamid, fluroxypyr, MCPB and pyroxsulam) had quantification rates between 20 and $80\,\%$, while six others (cloquintocet-mexyl, florasulam, halauxifen-
methyl, iodosulfuron-methyl-sodium, mefenpyr-diethyl and pinoxaden) were not quantified in any sample. With the exception



of clopyralid, fluroxypyr and mefenpyr-diethyl, these molecules have a persistence in soil of $5\,\mathrm{d}$ or less, explaining their rapid disappearance. Despite its short persistence in soil ($\mathrm{DT50_{soil}} = 3.5\,\mathrm{d}$) and medium application rate ($\mathrm{q} = 73\,\mathrm{\mu g\,kg^{-1}}$), fenpicoxamid was quantified in $100\,\%$ of the soil samples due to its high soil sorption ($\mathrm{K_{oc}} = 53233\,\mathrm{mL\,g^{-1}}$) and very low water solubility ($\mathrm{s} = 0.041\,\mathrm{mg\,L^{-1}}$). The low quantification rates of clopyralid and fluroxypyr in soil samples are probably due to
their high water solubility ($\mathrm{s} > 1000\,\mathrm{mg\,L^{-1}}$) and relatively high LQ in soil samples ($\mathrm{LQ} \geqslant 2.5\,\mathrm{\mu g\,kg^{-1}}$).

    By day 80, soil contents ranged from below $0.20\,\mathrm{\mu g\,kg^{-1}}$ (cloquintocet-mexyl and pinoxaden; lowest LQ) to $490 \pm_{\mathrm{sd}} 150\,\mathrm{\mu g\,kg^{-1}}$ (tebuconazole). This corresponded to residues from $0\,\%$ (no detection) to $120 \pm_{\Delta} 30\,\%$ (fluxapyroxad) of the initial mass (median $< 0.1\,\%$). The seven active substances quantified at a rate of $100\,\%$ on day 45 were still quantified in all samples on day 80, with the addition of MCPB (highest applied active substance). The remaining 10 active substances were quantified in no more
than $13\,\%$ of the samples. Compared to day 45, soil contents appeared to increase for five of the eight molecules systematically quantified above their LQ (fenpicoxamid, fluxapyroxad, mefentrifluconazole, MCPB and tebuconazole), particularly under the thin cover and the control; the observed increases ranged from $36 \pm_{\Delta} 48\,\%$ for fenpicoxamid to $220 \pm_{\Delta} 140\,\%$ for MCPB (excluding the thick cover on day 80 from the averages). These apparent increases even exceeded the initial mass applied (day –14) for fluxapyroxad, mefentrifluconazole and tebuconazole, reaching contents of $140 \pm_{\Delta} 20\,\%$, $120 \pm_{\Delta} 20\,\%$ and $110 \pm_{\Delta} 20\,\%$
of the initial mass, respectively. These molecules generally show the longest soil persistence ($\mathrm{DT50_{soil}} > 100\,\mathrm{d}$) and/or the highest soil sorption ($\mathrm{K_{oc}} > 4000\,\mathrm{mL\,g^{-1}}$) of all applied substances, with the exception of MCPB, whose presence in soil was renewed by the degradation of MCPA, of which it is a major metabolite. This apparent anomaly is likely due to differences in soil sampling procedures between the first two soil samplings (day 0 and day 45) and the third sampling (day 80). On day 80, the reduced soil mass sampled preferentially selected smaller aggregates, mainly from the topsoil where soil-adsorbed pesti-
cide contents are higher (rather than larger aggregates from the subsoil, which have lower soil-adsorbed pesticide contents). This introduced a bias that artificially increased the quantified contents of persistent, poorly soluble and/or soil-adsorbed pesticides compared to the more homogeneous samples of day 0 and day 45. As a result, the temporal trends observed in the soil compartment are likely biased; however, as sampling was consistent between modalities at each individual date, comparisons between modalities at a given date remain valid.

In comparison to our results, Silva et al. (2019), reported higher pesticide contents in agricultural topsoils collected in situ across Europe in 2015. These elevated contents are likely to be due to differences in study design: our soil samples were taken from an organic soil with a single pesticide application on day –14, whereas their study targeted conventional agricultural fields with recurrent pesticide use, selecting countries and crops with the highest pesticide application per hectare. As a result, they reported quantified residue contents as high as $2000\,\mathrm{\mu g\,kg^{-1}_{\mathrm{air\text{-}dried\ soil}}}$ (glyphosate) compared with our highest application rate of
$1450\,\mathrm{\mu g\,kg^{-1}}$ (MCPB). In addition, our study simulated cover crop conditions during a fallow period, with soil sampled under fully developed cover 94 days after the pesticide treatment (day 80); in contrast, samples of Silva et al. were collected between April and October, coinciding with the period of application of most pesticides. Pelletier and Agnan (2019) reported pesticide contents similar to ours in soil under maize cultivation, up to $270\,\mathrm{\mu g\,kg^{-1}_{\mathrm{dried\ soil}}}$ (S-metolachlor) eight days after application; these values are comparable to those observed in our study on day 0 (14 days after application), where contents reached a
maximum of $730\,\mathrm{\mu g\,kg^{-1}}$ (MCPA).





### 3.1.2 Soil solution concentration

By day 45, concentrations in soil solution samples ranged from below $0.025\,\mu g\,L^{-1}$ (halauxifen-methyl, lowest LQ) to $27\pm_{sd}13\,\mu g\,L^{-1}$ (clopyralid), corresponding to residues from $0\,\%$ (no detection) to $10\pm_{\Delta}5\,\%$ (clopyralid) of the initial mass (median $< 0.1\,\%$). Seven active substances (clopyralid, florasulam, fluroxypyr, fluxapyroxad, mesosulfuron-methyl, pyroxsulam and tebucona-
zole) were quantified in all samples. These molecules are characterised by high application rate ($q > 145\,\mu g\,kg^{-1}$), high leachability (GUS $> 2.8$) and/or high solubility ($s > 1000\,mg\,L^{-1}$). Four others (flonicamid, MCPA, MCPB and mefentri-fluconazole) had quantification rates between 7 and $93\,\%$, while five (cloquintocet-mexyl, halauxifen-methyl, iodosulfuron-methyl-sodium, mefenpyr-diethyl and pyraclostrobin) were not quantified in any sample. The non-detected substances are characterised by a persistence in soil of $5\,d$ or less, a low leachability (GUS $< 1.8$) and/or low solubility ($s < 10\,mg\,L^{-1}$).

By day 80, concentrations had dropped further from below $0.025\,\mu g\,L^{-1}$ (halauxifen-methyl, lowest LQ) to $9.9\pm_{sd}4.1\,\mu g\,L^{-1}$ (tebuconazole), corresponding to residues from $0\,\%$ (no detection) to $3\pm_{\Delta}5\,\%$ (mesosulfuron-methyl) of the initial mass (median $< 0.1\,\%$). Three of the seven active substances quantified at a rate of $100\,\%$ on day 45 (fluxapyroxad, mesosulfuron-methyl and tebuconazole) were still quantified in all samples on day 80. The other four are characterised by short soil persistence (DT50$_{soil} < 30\,d$) and high soil mobility ($K_{oc} < 75\,mL\,g^{-1}$), resulting in faster degradation and transfer out of the sampled
topsoil. Eight active substances (clopyralid, flonicamid, florasulam, fluroxypyr, MCPA, mefentrifluconazole, pyraclostrobin and pyroxsulam) were detected with rates between 13 and $80\,\%$, while five others (cloquintocet-mexyl, halauxifen-methyl, iodosulfuron-methyl-sodium, MCPB and mefenpyr-diethyl) were never detected, consistent with day 45 trends.

Compared to our results, Pelletier and Agnan (2019) reported similar pesticide concentrations in soil solution collected at a depth of $50\,cm$, with median values ranging from $0.01\,\mu g\,L^{-1}$ (2,4-D) to $5.20\,\mu g\,L^{-1}$ (S-metolachlor) over their four-year
maize field study (LQ from 0.01 to $0.60\,\mu g\,L^{-1}$). Similarly, Giuliano et al. (2021) observed maximum soil solution concentrations at $1\,m$ depth between $1.31\,\mu g\,L^{-1}$ (glyphosate) and $28.96\,\mu g\,L^{-1}$ (mesotrione) during their eight-year maize field study (LQ from 0.01 to $0.05\,\mu g\,L^{-1}$). In contrast, Vryzas et al. (2012) reported significantly higher concentrations, reaching up to $1166\,\mu g\,L^{-1}$ (atrazine) at $35\,cm$ depth in their four-year maize field study (LQ from 0.005 to $0.05\,\mu g\,L^{-1}$). This discrepancy can be attributed to preferential flow mechanisms facilitated by deep clay cracks in high clay soils under their semi-arid condi-
tions. Compared to these studies, our relatively high LQ (from 0.025 to $1.5\,\mu g\,L^{-1}$) limited our ability to follow all 18 active substances in the soil solution compartment.

### 3.1.3 Differences in compartments

To analyse both compartments simultaneously and to integrate data from all sampling dates, we performed a PCA on all quantification results (Fig. 2). The right panel of the figure shows the projection of each active substance on the first two dimensions
of the PCA. The first dimension, accounting for $60\,\%$ of the variance, separated the molecules in two groups: (1) negative values corresponded to substances such as mefentrifluconazole and tebuconazole, which have high soil sorption, high lipophilicity, low water solubility and/or long soil persistence; and (2) positive values corresponded to substances such as clopyralid or pyroxsulam, which have low soil sorption, low lipophilicity, high water solubility and/or short soil persistence. The second





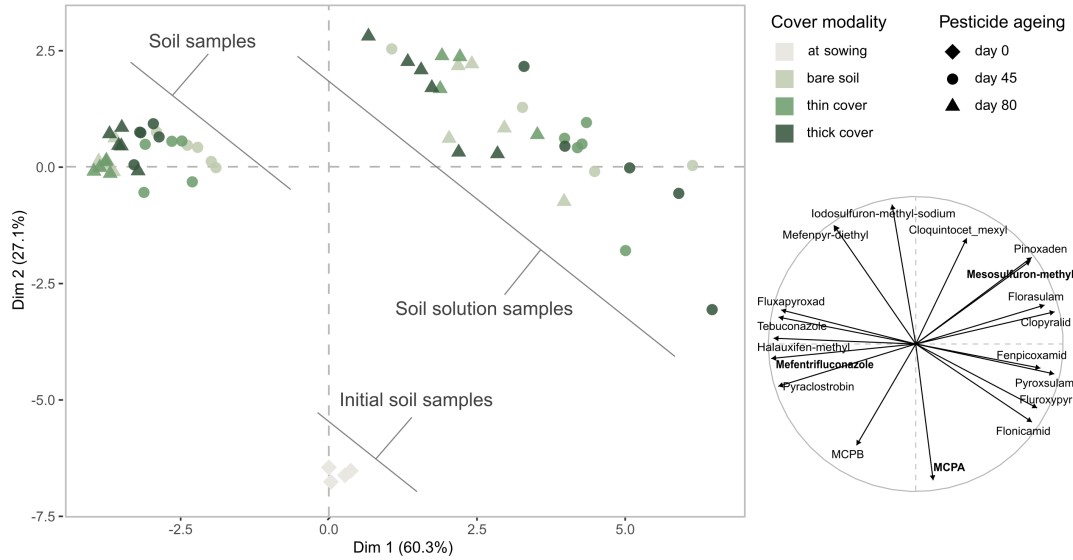

**Figure 2.** PCA of all sample quantifications: the relative distribution of active substances in soil and soil solution samples changed over time. Molecules in bold in the right panel are the three selected for further analysis in Sect. 3.2.

dimension, accounting for 27 % of the variance, further differentiated the active substances: (1) negative values corresponded
mainly to MCPA and MCPB, witch have highly application rates and low molecular masses while (2) positive values corre-
sponded to substances such as iodosulfuron-methyl-sodium and mesosulfuron-methyl, which have lower application rates and
higher molecular masses.

    Soil and soil solution samples were clearly separated by the first axis of the PCA, indicating that compartment was the main
contributor to variance. Initial soil samples (day 0) are concentrated on the negative side of the second dimension, dominated by
highly applied, low molecular mass molecules. Over time, the distribution of active substances changed in each compartment.
Soil samples initially moved to the upper left (day 45), reflecting a shift towards a dominance of molecules with higher soil
sorption, bioconcentration or persistence, before shifting further to the left (day 80). In contrast, soil solutions samples shifted
to the upper right (day 45), influenced by molecules with lower soil sorption, bioconcentration or persistence. By day 80, these
samples shifted up and left, reducing the influence of highly applied, low molecular mass molecules.

**3.2   Influence of cover types**

The temporal shifts analysed in the previous section highlight the dynamic speciation and redistribution of compounds within
each compartment. Given the observed influence of physicochemical properties on the temporal evolution of individual sub-
stances, we focus in this section on three contrasting molecules —mesosulfuron-methyl, MCPA and mefentrifluconazole (see
Supplement S3 for details of the selection process)— that were consistently detected in soil samples, examining their behaviour





under the different cover types. These compounds, in bold in the right panel of Fig. 2, serve as representative examples of the pesticides applied.

Mesosulfuron-methyl (a systemic post-emergence herbicide with moderate soil mobility, moderate water solubility, very low volatility and high molecular mass) showed uniform behaviour in all modalities in soil samples on day 45, with a soil content of ca. $3\,\mu g\,kg^{-1}$ (i.e. ca. $30\,\%$ of the initial applied mass on day –14; Fig. 3a). By day 80, the average content in soil

samples under the thin cover was $32\pm_\Delta 37\,\%$ higher than under the control ($p$-value $< 0.05$), whereas it was $37\pm_\Delta 22\,\%$ lower under the thick cover ($p$-value $< 0.05$). In the soil solution on day 45, the average concentration under the cover types was not different from the control, with a soil solution equivalent content of ca. $0.7\,\mu g\,kg^{-1}$ (i.e. ca. $7\,\%$ of the initial mass). However, by day 80, the average soil solution content under the thick cover had decreased by $58\pm_\Delta 14\,\%$ compared to the control ($p$-value $< 0.01$). The dual pattern observed in the soil samples for mesosulfuron-methyl —retention under the thin

cover and reduction under the thick cover— supports our two main hypotheses: (1) rhizofiltration driven by evapotranspiration and (2) enhanced biodegradation mediated by rhizospheric microorganisms.

(1) As the cover develops, the thin cover modifies soil water fluxes through evapotranspiration, a process that is likely to acts as rhizofiltration by retaining in the rhizosphere active substances that would otherwise leach deeper into the soil profile (Tarla et al., 2020). The higher contents under the thin cover crop therefore reflect a greater retention compared

to the leaching observed under the control, rather than an absolute increase in residue (Fig. 4, left and central panels). This is consistent with previous studies showing that cover crops increase soil permeability while decreasing drainage by removing soil moisture through evapotranspiration (Alletto, 2007; Unger and Vigil, 1998). However, this effect only became apparent 80 days after sowing, suggesting that it depends not only on the stage of development of the cover, but also on an adaptation period required to modify soil water fluxes and reverse initial leaching. While this effect was

evident in soil samples, it was not significant for mesosulfuron-methyl in soil solution samples under the thin cover on day 80 or the thick cover on day 45 (at equivalent biomass density of ca. $0.4\,t_{DM}\,ha^{-1}$).

(2) As the cover continues to grow and its root system develops, rhizospheric microbial activity increases, enhancing the biodegradation of pesticide residues (Cycoń et al., 2017; Eevers et al., 2017; McGuinness and Dowling, 2009). This process likely reduced the content of mesosulfuron-methyl in the soil under the thick cover compared to the control, as

biodegradation counteracted the increased retention of the cover (Fig. 4, right panel). This biodegradation probably acts in parallel to enzyme-driven catalysis from root exudates, fungi or other microorganisms, interaction with rhizospheric organic matter and plant uptake. As microbial abundance and diversity were not monitored and pesticide content in plant material (roots nor shoots) was not quantified, these mechanisms remain undifferentiated.

MCPA (a systemic post-emergence herbicide with high soil mobility, very high water solubility, low volatility and low

molecular mass) also showed a uniform behaviour in all modalities in soil samples on day 45, with a soil content of ca. $1.5\,\mu g\,kg^{-1}$ (i.e. ca. $0.25\,\%$ of the initial mass; Fig. 3b). By day 80, the average content in the soil samples under the thin cover was equivalent to that of the control, whereas it had decreased by $31\pm_\Delta 30\,\%$ under the thick cover ($p$-value $< 0.05$). In the soil solution, concentrations were at or below the LQ for all modalities on both dates, limiting further interpretation. Compared to mesosulfuron-methyl, average soil and soil solution equivalent contents were significantly lower for MCPA (below $0.5\,\%$ of




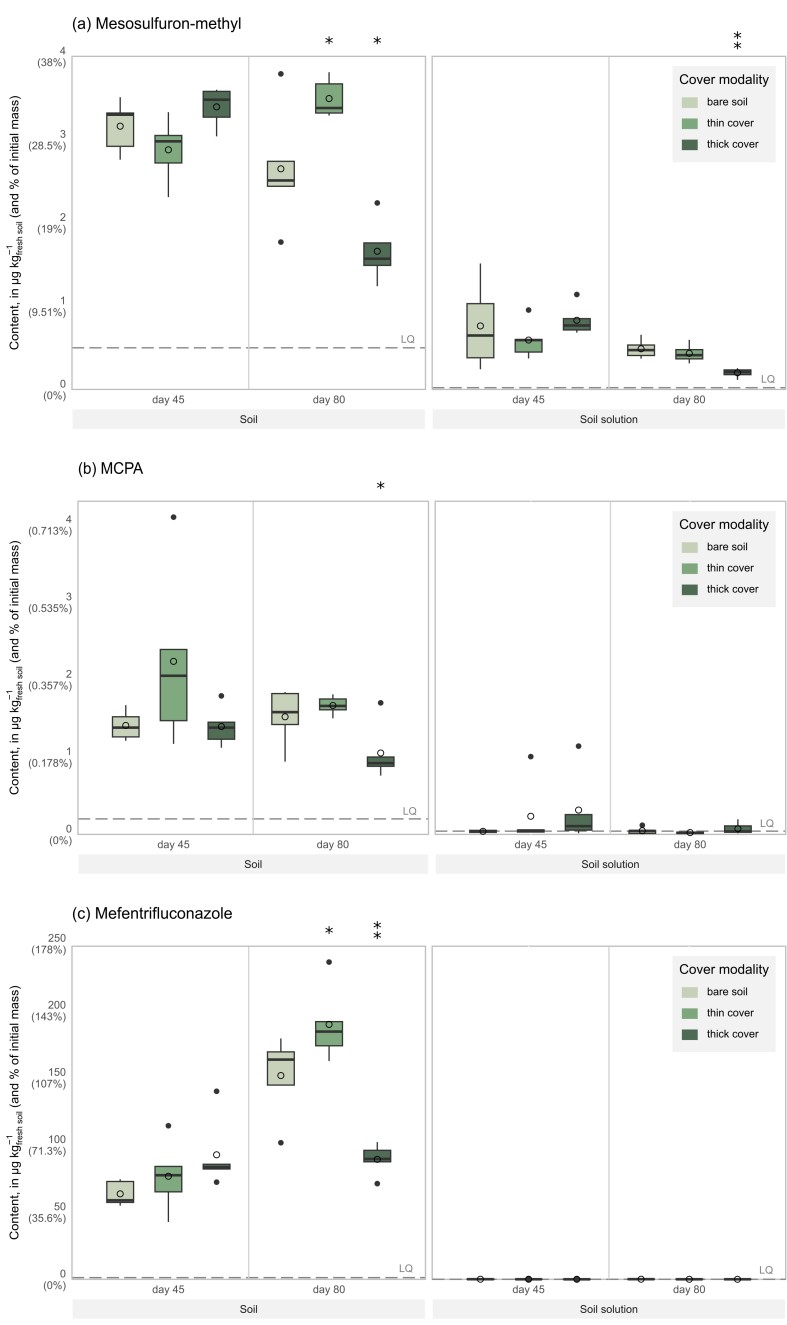

**Figure 3.** Active substance contents (in $\mu g\,kg^{-1}$ and in proportion of the initial applied mass on day $-14$) for three contrasting molecules (mesosulfuron-methyl, a; MCPA, b; mefentrifluconazole, c) under three cover modalities (bare soil, thin cover and thick cover) in two compartments (soil and soil solution) at two dates (day 45 and day 80). Stars above the graphs depict statistically significant unilateral differences between the cover types and the control (bare soil) at each date ($*$: $0.05 \geqslant p$-value $> 0.01$; $\overset{*}{*}$: $0.01 \geqslant p$-value $> 0.001$).





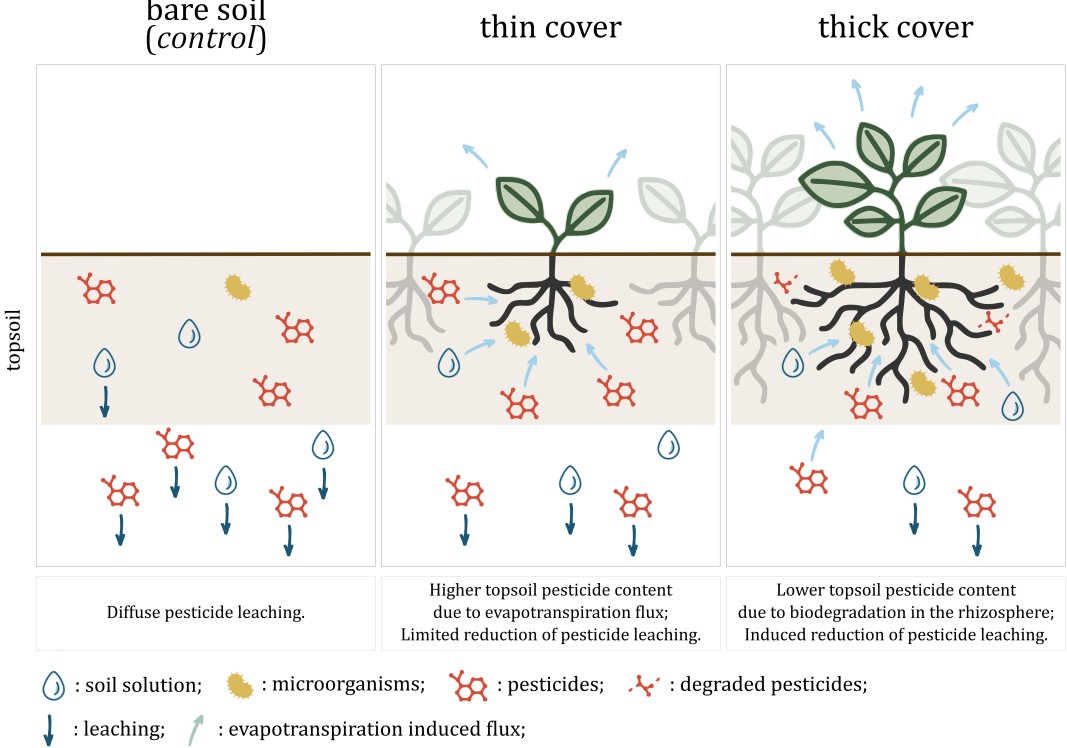

**Figure 4.** Cover crops reduce pesticide leaching by altering soil water fluxes through evapotranspiration and concentrating pesticides near roots where they are efficiently degraded by edaphic microbiota.

the initial mass), suggesting that a greater share of the initial mass was either transferred out of the system or degraded. The observed reduction in soil samples under the thick cover supports hypothesis (2), while the limited rhizofiltration effect under the thin cover (hypothesis 1) was likely due to the high soil mobility and very high solubility of MCPA.

Mefentrifluconazole (a systemic fungicide with very low soil mobility, low water solubility, low volatility and moderate molecular mass) also showed a uniform behaviour in all modalities in soil samples on day 45, with a soil content of ca. 75 µg kg$^{-1}$ (ca. 55 % of the initial mass; Fig. 3c). By day 80, the average content in the control soil samples had increased by $95 \pm_\Delta 19$ %, reaching ca. 150 µg kg$^{-1}$ (ca. 110 % of the initial mass), due to changes in soil sampling. On day 80, the average content in the soil samples under the thin cover was $25 \pm_\Delta 31$ % higher than under the control ($p$-value $< 0.05$), whereas it was $41 \pm_\Delta 14$ % lower under the thick cover ($p$-value $< 0.01$). As for MCPA, concentrations in the soil solution were at or below the LQ for all modalities on both dates. The results are, again, consistent with our hypotheses, showing (1) a significant increase in pesticide content under the thin cover compared to the control by day 80, driven by an evapotranspiration-induced rhizofiltration, and (2) a very significant decrease under the thick cover, likely due to biodegradation facilitated by microorganisms stimulated by the developed rhizosphere.



In summary, the dual pattern of pesticide retention under the thin cover and degradation under the thick cover was particularly evident after 80 days, when the root system of the cover crops had developed sufficiently. This was mainly observed

in soil samples, where pesticide contents were higher than in soil solution. In soil solution samples, the effect was detectable at concentrations above the LQ, with only a few statistically significant differences between the cover types and the control, warranting further investigation. In this study, a biomass of at least $1.12 \pm_{\mathrm{sd}} 0.02 \pm_{\Delta} 0.18 \, \mathrm{t_{DM} \, ha^{-1}}$ was required to achieve a significant reduction of the active substances in both soil and soil solution by day 80. This threshold is lower than the $2 \, \mathrm{t_{DM} \, ha^{-1}}$ biomass reported by Alletto et al. (2012) as necessary to observe similar effect in field experiments. Note that our

thin and thick covers are composed of different species: species-specific characteristics beyond growth rate and root density may influence these effects. The observed patterns were consistent for all three molecules studied, despite their contrasting physicochemical properties, suggesting that these effects may be generalised to other active substances, with varying magnitudes (see figures in Supplement S5 for similar graphs of the other active substances). The magnitude of the effect correlated with soil mobility and water solubility, suggesting that the properties of the compounds may help predict whether cover crops

will significantly alter their fate in soil.

### 3.3 Physicochemical properties

Building on the previous results, this section examines the relationship between the physicochemical properties of the applied substances and the differences between their soil content under both cover types and the control on day 80. Although only eight substances showed quantified soil contents on day 80, analysis of individual physicochemical trends provide insights

into the processes influencing the interaction between soil covers and active substance behaviours. Specifically, we examined four physicochemical properties —soil mobility ($K_{\mathrm{oc}}$), water solubility (s), molecular mass (m) and volatility (p)— which correspond to persistence in soil, transfer to soil solution, tendency for plant uptake and transfer to air, respectively (Fig. 5). In general, the deviation from the control (i.e. the absolute value of the difference in content $|\Delta C|$) increased with higher $K_{\mathrm{oc}}$ (Fig. 5a) and higher molecular mass (Fig. 5c), whereas it decreased with higher water solubility (Fig. 5b) and higher vapour

pressure (Fig. 5d).

For soil mobility, the soil sorption coefficients for the 18 applied active substances ranged from $1.6$ (flonicamid) to $53000 \, \mathrm{mL \, g^{-1}}$ (fenpicoxamid) and substances quantified by day 80 had sorption coefficients above $K_{\mathrm{oc}} \geqslant 74 \, \mathrm{mL \, g^{-1}}$ (MCPA). By day 80, the most mobile substances had been transferred out of the soil or degraded, limiting the effect the cover crops could have on them. A linear fit, with its $90\%$ confidence interval, of the deviation from the control under the thick cover ($R^2 = 0.68$,

$p$-value $< 0.05$; Fig. 5a) indicated that active substances with soil sorption coefficient greater than $K_{\mathrm{oc}} \geqslant 160 \pm_{\Delta}{}^{1700}_{150} \, \mathrm{mL \, g^{-1}}$ experienced a reduction in soil content of at least $33\%$. Higher soil sorption ensured lower mobility and longer retention of the substances within the microbiologically active rhizosphere, allowing the effects of the thick cover to fully manifest. While sorbed substances are typically less bioavailable, higher soil organic matter from root systems and exudates can both enhance pesticide adsorption and facilitate desorption. This dual process can enhance biodegradation by supporting microorganisms in

soils with high organic matter content, enabling them to break down pesticides more efficiently (Eevers et al., 2017).





**Figure 5.** Differences in active substance soil contents compared to the control (bare soil) on day 80, for the eight active substance with $100\%$ quantification rate and for both cover types, in function of the active substance's: (a) soil mobility (as $\log(K_{oc})$), (b) water solubility (as $\log(s)$), (c) molecular mass (m) and (d) volatility (as $\log(p)$). The coloured lines represent linear fits for both cover types, with $90\%$ confidence intervals. The three contrasting molecules are tagged with a letter below them (mesosulfuron-methyl: a; MCPA: b; mefentrifluconazole: c). Stars above the error bars depict statistically significant unilateral differences between the cover type and the control at each date ($*$: $0.05 \geqslant p$-value $> 0.01$; $\overset{*}{*}$: $0.01 \geqslant p$-value $> 0.001$).



Water solubility of the 18 studied active substances ranged from $0.041$ (fenpicoxamid) to $250000\,\mathrm{mg\,L^{-1}}$ (MCPA), with this range being largely observed up to day 80. A linear fit ($R^2 = 0.49$, $p$-value $\simeq 0.05$; Fig. 5b) indicated that substances with solubility under $\mathrm{s} \leqslant 1400 \pm_\Delta{}^{61000}_{1400}\,\mathrm{mg\,L^{-1}}$ had their soil content reduced by at least $33\,\%$ under the thick cover. More soluble compounds leached more rapidly outside of the rhizosphere, reducing the effect of the cover on their soil content.

Relative molecular mass of the studied active substances ranged from $190$ (clopyralid) to $620$ (fenpicoxamid) and substances quantified by day 80 had molecular mass above $\mathrm{m} \geqslant 200$ (MCPA). A linear fit ($R^2 = 0.68$, $p$-value $< 0.05$; Fig. 5c) indicated that substances with molecular mass above $\mathrm{m} \geqslant 280 \pm_\Delta 140$ had their soil content reduced by at least $33\,\%$ under the thick cover. However, the 18 molecules analysed in this study show a general inverse relationship between molecular mass and solubility. This suggests that compounds with lower molecular mass may be less degraded due to increased leaching and that the observed results might not reflect the intrinsic effects of molecular mass. This would explain the discrepancy with some existing literature, such as that reported by Jia et al. (2023).

For volatility, the vapour pressure of the studied substances ranged from $3.5\,10^{-9}$ (mesosulfuron-methyl) to $1.4\,\mathrm{mPa}$ (clopyralid), with substances quantified up to day 80 having vapour pressures less than $\mathrm{p} \leqslant 0.4\,\mathrm{mPa}$ (MCPA). A linear fit ($R^2 = 0.60$, $p$-value $< 0.05$; Fig. 5d) indicated that substances with vapour pressures greater than $\mathrm{p} \geqslant 1.3\,10^{-4} \pm_\Delta{}^{1.2\,10^{-2}}_{1.2\,10^{-4}}\,\mathrm{mPa}$ had their soil content increased by less than $20\,\%$ higher under the thin cover, suggestion that volatilisation resulted in a greater loss of soil content before the cover crop could take effect. While the cover still had an effect on the more volatile substances, it was less pronounced that for the less volatile molecules.

While most deviations from the control in soil samples under the thick cover were significantly different from zero on day 80, differences under the thin cover or in soil solution samples were generally not statistically significant. The same pattern was observed at day 45. While this may suggest a lack of effect of the cover crops at lower biomass or earlier time, it could also be due to insufficient statistical power in the experimental setup. To guide further experimental design, we calculated the minimum sample sizes required to achieve at least $80\,\%$ statistical power under similar conditions of active substance levels, variances between independent replicates and cover developments (see Supplement S4 for details). For soil samples, adequate statistical power was already achieved on day 80 with five replicates (except for MCPA, which required eight replicates); however, for soil solution samples, a median sample size of 14 replicates was required (with a maximum of 118 for tebuconazole; see Table S8 in the Supplement).

In conclusion, cover crops affect the presence of active substances in the soil over a wide range of physicochemical properties, as highlighted by the non-zero deviation from the control for both cover types and all quantified substances on day 80 in the soil samples. Our results suggest that even persistent or adsorbed pesticides continue to be degraded as long as cover crops are maintained. Under the thick cover, substances with moderate to non-mobility in soil ($\mathrm{K_{oc}} \geqslant 160\,\mathrm{mL\,g^{-1}}$), low to high water solubility ($\mathrm{s} \leqslant 1400\,\mathrm{mg\,L^{-1}}$) and/or moderate to high molecular mass ($\mathrm{m} \geqslant 280$) experienced at leas a $33\,\%$ reduction in soil content by day 80, compared to the control (where leaching occurred). As a result of the biodegradation of evapotranspiration-retained residue in the topsoil, the cover also reduced the leaching of the active substances. In Wallonia (southern half of Belgium), 141 authorised active substances —including $30\,\%$ of the most frequently used active substances in the period 2015–2020— fall within all three thresholds and mainly concern potatoes, sugar beet and winter cereals (with avail-





ability of PPDB data in May 2024; data extracted from Corder, 2023, and *phytoweb.be* in November 2024). The adoption of dense cover crops during the fallow period could therefore play a important role in reducing pesticide leaching to groundwater.

## 3.4 Agronomic interest

The results of the previous sections show that cover crops can significantly reduce the environmental impact of pesticides by de-
creasing their presence in the soil and limiting their transfer to groundwater. While pesticide concentration in soil solution may appear negligible compared to soil content, cumulative leaching can lead to significant groundwater contamination, particularly during aquifer recharge periods. The observed reductions in pesticide levels highlight the potential of cover crops to protect water quality. Although this effect may not be sufficient for highly volatile and soluble pesticides, it represents an important step in phytoremediation. Unlike long-term strategies such as multi-year miscanthus (*Miscanthus × giganteus*) plantations for
trace metal remediation or soil excavation, cover crops provide a flexible approach without limiting field availability.

As the effects of cover crops on pesticide dynamics only become apparent after a period of growth and adaptation, cover crops should be established as soon as possible after harvest to maximise pesticide reduction. In regions where the fallow period coincides with the rainy season, early sowing allows the cover crop to reach a sufficient growth stage to reduce pesticide levels before rainfalls initiate groundwater recharge. This is particularly important for highly soluble compounds, which migrate
rapidly beyond the rhizosphere once soils reach field capacity. In regions where the fallow period occurs in winter, temperature is also a key factor, as microbial degradation rates decrease in colder conditions. Cover crops help to maintain warmer soil temperatures in autumn and buffer temperature fluctuations (Imfeld et al., 2024; Patton et al., 2024), thereby extending the window for microbial degradation before winter dormancy. However, these findings apply not only to temperate regions with groundwater recharge periods driven by winter rainfall, but also to subtropical regions where fallow periods coincide with
summer rainfall (Potter et al., 2007).

Cover crops influence soil microbial dynamics by altering microbial abundance, activity and diversity, thereby increasing the biodegradation of pesticide residues. However, this increased degradation should not be used as a justification for maintaining or increasing pesticide use as numerous studies have shown that pesticide use can negatively affect soil microbial communities, altering microbial diversity and enzymatic activity in soils (Chowdhury et al., 2008; Cycoń et al., 2017; Das et al., 2016). In
addition, pesticide residues can directly inhibit the establishment of subsequent crops, including cover crops, thereby reducing biomass production and transpiration rates (Feng et al., 2024; Palhano et al., 2018; Rector et al., 2020; Silva, 2023). Therefore, cover crops should be integrated into broader agroecological strategies, such as integrated pest management (IPM), to reduce reliance on pesticides and increase ecosystem resilience. Prioritising non-chemical methods for cover crop termination is also essential to avoid introducing new pesticide residues into the soil.
The efficiency of phytoremediation depends on both the botanical family of the cover crop and the microbial strains present in the soil (Hussain et al., 2009; Jia et al., 2023; Wojciechowski et al., 2023). Certain plant species are more effective than others at retaining or degrading specific pesticide compounds, with annuals often showing higher remediation efficiencies than perennials due to their rapid biomass growth and high transpiration rates (Jia et al., 2023). Our results suggest that cover crops





can reduce pesticide residues across a broad range of molecules and that choosing fast-growing species with dense root systems

can further enhance their remediation potential, as has also been observed in weed management (MacLaren et al., 2019).

In addition to their role in phytoremediation, cover crops also affect the fate of pesticides through processes not investigated in this study, such as plant uptake. Pesticide translocation within plants depends on physicochemical properties such as lipophilicity ($K_{ow}$), water solubility and molecular mass. Although accumulation is generally greater in roots (Chuluun et al., 2009), compounds with $K_{ow}$ values between 1 and 3 can be transported from roots to shoots (Jia et al., 2023). Although this pa-

per does not address the ultimate fate of pesticide-contaminated biomass, the risk of hazardous pesticide residues accumulating in cover crops is likely to be minimal if the preceding crop was considered safe for food or feed. However, a notable exception concerns late-flowering cover crops that could provide contaminated floral resources for pollinators following a non-flowering main crop (for which pesticide application posed no risk to pollinators; Morrison et al., 2023; Sanchez-Bayo and Goka, 2014; Zioga et al., 2023). In such cases, selection of non-flowering covers or topping before flowering may help to reduce risks.

Finally, any practice that increases microbial activity will contribute to pesticide degradation. Crop diversification, vegetative buffers or permanent cover all promote a more active soil microbiota, thereby facilitating pesticide degradation and (directly or indirectly) reducing leaching (Krutz et al., 2006; Venter et al., 2016). This approach could be particularly relevant for plots transitioning to organic farming, accelerating the reduction of pesticide residues in the soil. Cover crops also play a critical role in reducing erosion-related pesticide runoff, making them valuable in protecting both surface and groundwater quality

from pesticide contamination. By acting directly in the soil compartment where pesticides are applied, rather than treating groundwater at the point of extraction, such measures also help to reduce pesticide contamination in other environmental compartments. This can directly improve water quality, and it is conceivable that agri-environmental subsidies for long-term, dense cover crops could be partly funded through drinking water tariffs, as this practice reduces downstream costs associated with water remediation and sanitation.

## 3.5 Limitations and perspectives

This study provides valuable insights into the role of cover crops in pesticide fate and persistence, but has several limitations.

Our main hypothesis highlighted the role of microorganisms in pesticide biodegradation, but we were unable to directly monitor microbial activity. Further research integrating both pesticide quantification and microbial activity measurements would provide valuable confirmation of this hypothesis. Similarly, although we tested two different types of cover crops, their

different growth patterns led us to asses cover density rather than the specific effects of cover species. Further experiments comparing single and multi-species covers at different densities would improve our understanding of these processes.

Metabolites can be more toxic and persistent than parent compounds, and biodegradation typically involves successive transformations —oxidation, reduction, hydrolysis, conjugation or polymerization— which further influence persistence (Fenner et al., 2013; Tixier et al., 2000, 2002). The lack of their analysis is a key limitation of our study. For example, mefentriflucona-

zole produces trifluoroacetate (TFA), as highly persistent polyfluorinated metabolite, raising concerns about drinking water contamination by per- and poly-fluoroalkyl substances (PFAS) in Europe (Burtscher-Schaden et al., 2024; Joerss et al., 2024; Freeling and Björnsdotter, 2023; PAN Europe and Générations Futures, 2023). While our results suggest that cover crops ac-





celerate the degradation of mefentrifluconazole, the fate of its metabolites remains uncertain. Future research should therefore include these metabolites and evaluate the role of co-formulants to better understand degradation dynamics.

Although greenhouse experiments cannot fully replicate field conditions, mesoscale setups are relevant for studying pesticide fate and ecotoxicological effects (Chaplin-Kramer et al., 2019). In our study, 10 L pots allowed controlled assessments but limited leaching assessments due to the shallow soil depth. The inability to collect soil solutions at multiple depths highlights the need for field validation, as deeper soil profiles may influence observed effects such as increased leaching or resurgence of residues from lower horizons due to evapotranspiration-induced water fluxes (Pelletier and Agnan, 2019). In addition, root

channels and earthworm burrows enhance microbial degradation (Mallawatantri et al., 1996), but also create preferential flow paths that may accelerate pesticide transport beyond microbial activity zones. A better understanding of the vertical transfer dynamics, runoff and temporal concentration variations is essential to assess the cover crop ecosystem service of groundwater pollution mitigation. Furthermore, while our controlled experiment isolated soil effects, variations in soil properties (e.g. pH, organic matter content) and environmental factors (e.g. temperature, rainfall, field heterogeneity) are likely to influence

pesticide behaviour in the field.

    While our study assessed pesticide persistence using a linear framework based on individual physicochemical properties, we acknowledge that complex interactions between pesticides and other contaminants may introduce non-linear effects. Furthermore, our approach focused on generalisable trends and did not take into account the molecular specificity of individual active substances, although structural features such as aromatic rings and halogen atoms (e.g. chlorine, fluorine) have a strong

influence on pesticide persistence and biodegradability (Calvet et al., 2005; Naumann, 2000).

    To refine our understanding of pesticide retention and degradation mechanisms under different cover conditions, future research should prioritise the following key areas:

(1) direct measurement of soil microbial biomass and activity to better characterise microbial interaction with the cover and contributions to pesticide degradation;

(2) systematic assessment of pesticide metabolite to evaluate their persistence and potential ecological impact;

(3) lowering the LQ in soil solution analyses to improve interpretation and allow more accurate tracking of pesticide concentrations in soil solution and leaching potential. This requires increased sampling volumes, either by using additional rhizons in field settings or by installing full-scale lysimeters;

(4) increasing sampling frequency to refine degradation kinetics and establish biomass thresholds relevant to pesticide degra-

dation;

(5) testing different cover crop species and densities to precise needed specifications. Multi year field trials under different climatic conditions, as well as multi-site trials with different pedoclimatic and microbiota conditions would provide a more comprehensive assessment. Control treatments with cover crops grown on untreated soils would help to isolate the effects of pesticide residues on biomass production and evapotranspiration;

(6) investigate pesticide uptake by cover crops to complete mass balance assessments and evaluate potential risks, including exposure pathways for pollinators.

    Addressing these limitations will improve our understanding of the influence of cover crops on the fate of pesticide residues in the soil and help support more sustainable agricultural management practices.



## 4    Conclusions

Reducing pesticide use —through improved application techniques, pest pressure management and agricultural system re-
design— is the primary strategy for mitigating pesticide-related environmental externalities and protecting surface and ground-
water quality. Beyond reduction strategies, ecological engineering offers complementary solutions to limit pesticide transfer
from agricultural soils to water. In this paper, we investigated the influence of newly sown cover crops on soil pesticide residues
from previous growing seasons by comparing pesticide levels in soil and soil solution over a three months greenhouse experi-
ment under three modalities: a thin cover, a thick cover and a control (bare soil; Fig. 1).

Our results show that living cover crops enhance the degradation of pesticide residues in soil and soil solution, supporting
their use as a remediation strategy for a wide range of pesticide molecules. Furthermore, our results provide a categorisa-
tion of pesticides influenced by cover crops: well-developed living cover crops 80 days after sowing with a biomass of more
than $1\,\mathrm{t_{DM}\,ha^{-1}}$ significantly reduced soil residue contents by at least $33\,\%$ for compounds with low to high water solu-
bility ($\mathrm{s} \leqslant 1400\,\mathrm{mg\,L^{-1}}$) and low to moderate soil mobility ($\mathrm{K_{oc}} \geqslant 160\,\mathrm{mL\,g^{-1}}$). In Wallonia, $30\,\%$ of the most frequently
used active substances fall within these thresholds, mainly concerning potatoes, sugar beet and winter cereals. These results
confirm previous results on individual compounds, cover crop types and soil compartments, while introducing thresholds for
physicochemical properties associated with significant pesticide degradation.

The proposed mechanism of pesticide residue degradation by cover crop builds on existing literature. We hypothesise that
cover crops reduce pesticide leaching by altering soil water fluxes though evapotranspiration and by concentrating pesticides
near the roots, thereby prolonging their residence in the microbiologically active rhizosphere where biodegradation is enhanced
(Fig. 4). The observed reduction in pesticide soil content is likely to be driven by edaphic microorganisms, as cover crops
promote biodegradation by stimulating native soil microbiota, rather than direct uptake by plants.

By acting directly in the soil where pesticides are applied, cover crops limit pesticide transfers to other environmental
compartments, particularly groundwater. As pesticide degradation is carried out by diverse microbial communities, these results
highlight the importance of maintaining biologically active soils. They also highlight the need to carefully consider the critical
transition period between crop harvest and cover crop establishment, as reduced evapotranspiration can increase pesticide
leaching before the cover crop is fully developed. This underlines the importance of sowing cover crops as soon as possible
after harvest to maximise their impact on pesticide residues, as their effect only becomes apparent after a period of growth and
adaptation. These findings also reinforce the need to reduce the overall use of pesticides, as they can have a negative impact
on soil microbial diversity. Integrating cover crops into broader agroecological strategies, such as IPM, offers a promising
approach to reducing reliance on pesticide while increasing the resilience of agroecosystems.

*Data availability.*    All raw data are available in the Supplement.



*Author contributions.* **NV:** Funding Acquisition, Conceptualization, Investigation (equal), Data Curation, Formal Analysis, Visualization,
Writing – Original Draft Preparation, Writing – Review & Editing. **IT:** Investigation (equal), Writing – Review & Editing (supporting). **AB:**
Formal Analysis (pesticide quantification), Writing – Review & Editing (supporting). **YA:** Supervision, Funding Acquisition, Conceptual-
ization, Investigation (supporting), Writing – Review & Editing.

*Competing interests.* The authors declare that they have no conflict of interest.

*Acknowledgements.* The authors express their gratitude to the Baillet Latour Fund, the Belgian Province of Luxembourg and the CER Groupe
for their financial support. They also thank Huges Falys and the UCLouvain University Farms for their time and for providing the soil for
the experiment, as well as Marc Migon and the UCLouvain Plant Cultivation Facilities (SEFY) for their time and for providing greenhouse
space. Additionally, they are grateful to Frédéric Brodkom and the UCLouvain Science and Technology Library for their time and resources
in supporting the open-access article processing charges. NV would like to extend special thanks to Éric Vandevoorde for his assistance in
managing the *phytoweb.be* database, Céline Chevalier for her graphic support, and Basile Herpigny, Océane Duluins and Philippe V. Baret
for their constructive reviews.

*Financial support.* This research was financially supported by the Baillet Latour Fund, the Belgian Province of Luxembourg, the CER
Groupe and the UCLouvain Science and Technology Library.



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
