# Peer review of "Living cover crops alter the fate of pesticide residues in soil: influence of pesticide physicochemical properties"

_EGUsphere, 2025_

## Author Response (AR1)

**Author's response**

Living cover crops reduce pesticide residues in agricultural soil

Reviewer comments are integrated in red. Our responses and proposed changes are included in black. Line numbers refer to the initial manuscript.

**Table of contents**

**1) Response to RC1 (Eglantina Lopez Echartea)**

**a) General comments**

This manuscript presents an investigation into the influence of cover crops on the fate of multiple pesticides compounds in soil and soil solution under pots and greenhouse conditions. The experimental design is innovative in applying a broad-spectrum pesticide mixture and monitoring its dynamics over time using analytical techniques. The PCA and physicochemical profiling contribute to a better understanding of how chemical properties influence persistence and distribution. However, the study makes several claims, particularly around microbial degradation and rhizosphere effects that are not fully supported by direct measurements. While the authors acknowledge some limitations, further caution is required in the interpretation of mechanisms. Overall, the manuscript provides valuable data but needs significant revisions in terms of how conclusions are framed.

Thank you for your thoughtful and constructive comments. Please find below our detailed responses. As it was requested that a revised manuscript should not be submitted with our response, the proposed changes are described with reference to line numbers from the preprint version of the manuscript.

**b) Specific comments**

**i) Scientific significance**

The study offers useful insight into pesticide-soil interactions and the potential of cover crops to affect pesticide retention and degradation. However, its broader contribution is limited by the absence of key complementary measurements (e.g., microbial activity, metabolite formation, evapotranspiration, accumulation in plants). These gaps limit the mechanistic depth needed to substantially advance the field.

We appreciate your emphasis on the importance of complementary measurements to deepen the mechanistic understanding of pesticide–soil interactions.

As stated in lines 92–94, we hypothesised based on existing literature that pesticide accumulation in plants over an 80-day period plays a lesser role than soil degradation, justifying the exclusion of plant tissue analysis in this study. We propose to reclarify this in line 466 by adding: "*since plant uptake generally plays a smaller role in pesticide dissipation than soil degradation (Tarla et al., 2020)".* While we recognise that this limits the mechanistic depth of our findings (as acknowledged in lines 525–526), this decision was also influenced by practical constraints, including limited laboratory capacity to analyse an additional matrix and financial limitations on the number of analyses that could be performed.

The same applies to microbial activity monitoring (explicitly acknowledged as a primary limitation in lines 482–484 and 512–513) and metabolite quantification (mentioned in line 155 and further discussed in lines 486–494 and 515).

We hope that the preliminary results presented in our paper will help to support future research efforts and funding aimed at broadening the analytical scope. Ideally, this would include the quantification of a wider range of active substances and their metabolites, the monitoring of plant uptake and the microbial dynamics, and the use of an experimental set-up that takes into account for both leaching and evapotranspiration processes.

**ii) Interpretation of results**

- Several conclusions suggest microbial degradation or rhizospheric effects (e.g., Fig. 4 and the discussion around "efficient degradation"), yet no microbial, enzymatic, or metabolite data are provided. These should be clearly framed as hypotheses rather than findings.
- The conclusion that increased pesticide content under thin cover crops reflects rhizofiltration (rather than slower degradation or less mobility) is speculative without additional data on leaching or degradation pathways.

To further clarify that microbiological monitoring was not included in the analysis, we propose to revise lines 94–96 into: "*As our objective was to identify trends in the physicochemical properties*

*of the active substances affected by cover crops, we did not include microbiological monitoring in our analysis*."

To mark a clearer distinction between our results and the hypothesised mechanism, we propose in section 3.2 to move the results and discussion on the behaviour of the three contrasted molecules (lines 312–321, 339–347, and 348–357, as well as Figure 3) to the Supplement and to clearly rename the section into "*Hypothesised mechanism*". See also our response to your comment on Figure 4 and the conceptual model.

To accommodate these modifications, we propose to modify lines 306–311 into: "*The shifts analysed in the previous section highlight the dynamic speciation and redistribution of compounds within each soil compartment over time. PERMANOVA results showed that, after soil compartments and sampling dates, cover modalities were the third most statistically significant factor explaining the variability in pesticide content between samples. Focusing on soil samples, the evolution of pesticide content over time and between cover modalities —detailed in Supplements S3 and S4— showed a dual trend after 80 days: (1) higher retention under thin cover (relative to thick cover and control), and (2) greater reduction under thick cover (relative to thin cover and control). These patterns support our two main hypotheses: (1) that rhizofiltration, driven by evapotranspiration, contributes to pesticide retention under less developed covers, and (2) that enhanced microbial biodegradation under thicker, more developed covers drives pesticide degradation. This leads to the following hypothesised mechanism:*". See also our response to your comment on the statistical testing.

**iii) Soil and system properties**

The study does not monitor soil physicochemical parameters over time (e.g., pH, organic matter, nutrients, microbial biomass), which are known to influence pesticide dynamics and could be differentially affected by the cover modalities. Their omission limits confidence in treatment effect attribution.

We agree that soil physicochemical parameters —such as pH, organic matter, nutrient content, and microbial biomass— can influence pesticide dynamics and may themselves be influenced by cover crops. Differential changes in these parameters could contribute to the observed differences in pesticide dynamics between modalities, and their absence limits the ability to confidently attribute observed effects to specific mechanisms.

However, all modalities were conducted on the same homogenised soil, and the 80-day period of cover crop growth is unlikely to be sufficient for significant divergence in bulk soil properties. While cover crops are known to affect these parameters over time —particularly after termination with incorporation, which is outside the scope of our study—, existing literature suggests that significant changes in bulk soil properties typically require several years of cover cropping (Blanco-Canqui et al., 2023; Wang et al., 2020). We propose to precise this by adding the following after line 153: "*As all modalities were conducted on the same homogenised soil, and given that significant changes in bulk soil properties generally require several years of cover cropping (Blanco-Canqui et al., 2023; Wang et al., 2020), we considered the 80-day cover crop growth period insufficient to induce meaningful divergence in soil physicochemical parameters (e.g. pH, organic matter, nutrients). Consequently, these parameters were not monitored beyond the initial soil characterisation.*"

We acknowledge that localised rhizosphere scale effects do occur and that their monitoring could help increase the confidence in the proposed hypothesised mechanism. However, investigating such micro-scale mechanisms was beyond our capabilities at the time of the experiment.

We agree that future studies incorporating temporal monitoring of soil properties and microbial dynamics would be valuable to better disentangle the mechanisms underlying observed effects, as noted in lines 503–505 and 521–523 of the manuscript.

**iv) Statistical testing**

The PCA in Figure 2 provides an summary of variance but lacks statistical support. Consider adding a PERMANOVA or similar test to assess whether observed groupings (e.g., by compartment or time) are statistically significant.

We appreciate this insightful suggestion and have addressed it by conducting a PERMANOVA to statistically support the groupings observed in the PCA. We propose the following modifications to the MS:

- Addition at the end of line 194: "*Permutational multivariate analyses of variance (PERMANOVA) were performed on the PCA to discuss results, using the* R *function* `vegan::adonis2` *(Oksanen et al., 2025). The homogeneity of the multivariate dispersion between the analysed groups was confirmed (*p*-value > 0.52), supporting the robustness of the observed patterns.*"

- Addition after line 304: "*These visual patterns were statistically supported by PERMANOVA, which demonstrated that soil compartment, sampling date, and cover modality each independently and significantly influenced the distribution of active substance levels. Compartment alone accounted for 68.5 % of the variance (*p*-value < 0.001), while date and modality explained 19.4 % (*p*-value < 0.001) and 16.0 % (*p*-value < 0.01), respectively. Combined, these three factors explained 88.3 % of the variance, increasing to 91.5 % when interactions were included. These results confirm that the separation observed in PCA space reflects differentiated trajectories of active substance evolution across soil compartments, sampling times and cover modalities.*"

See also the proposed change to the next paragraph (lines 306-311) in our response to your comment on interpretation of results.

**v) Relevance of application rates**

Please clarify how the pesticide application compares to realistic field conditions. Are the dosages representative of actual agricultural use, or are they higher for analytical purposes?

As stated in line 113, we applied pesticides at their maximum authorised dose, as there are no official 'standard doses' for pesticide use in Belgium. This may in some cases overestimate application rates compared to typical field practices. However, we used soil from a certified organic plot (see line 107) to minimise background pesticide contamination. This choice may in turn lead to an underestimation of residue levels found in conventional soils. We believe that the combination of these potential over- and underestimations results in pesticide residue levels that are representative of average field conditions. This interpretation is supported by comparison with findings from other studies in the literature (see lines 255–265 and 283–291).

**vi) Metabolite monitoring**

The lack of metabolite analysis is a critical limitation. Many pesticide degradation products can be more toxic or persistent than their parent compounds. This should be discussed in more detail and noted explicitly in the conclusions.

We acknowledge that the lack of metabolite quantification is a critical limitation of our study, as you pointed out. This issue is raised in line 155 and discussed in more detail in lines 486–494 and 515, particularly in relation to the potentially greater toxicity or persistence of some metabolites compared to their parent compounds.

At the time of the study, we faced analytical and financial constraints that limited our ability to systematically include metabolites in the analysis. Given these constraints, we chose to focus on a broad range of active substances to capture a variety of physicochemical behaviours, rather than conducting a more targeted and comprehensive tracking of metabolites of each substance. While this approach prevents us from drawing conclusions about degradation pathways, it does not affect the observed patterns of pesticide distribution and retention, nor the established thresholds of physicochemical properties of degraded active substances and cover densities to achieve significant reductions in pesticide levels.

In response to your suggestion, we propose to make this limitation explicit in the conclusion by adding the following sentence at the end of line 548: "*Major limitations of this study include the lack of direct measurements of soil microbial biomass and activity, and the lack of systematic assessment of pesticide metabolites.*"

**vii) Figure 4 and conceptual model**

The caption and narrative accompanying Figure 4 suggest definitive degradation by microbiota, which is not demonstrated in the study. This should be revised to reflect a proposed mechanism, unless additional data are included.

Thank you for pointing out the need to clearly distinguish between our observed results and the hypothesised mechanism. To address this, we propose the following revisions:

- **Restructuring Section 3.2**: We propose to move the results and discussion regarding the behaviour of the three contrasted molecules (lines 312–321, 339–347, and 348–357, as well as Figure 3) to the Supplement and to retitle the section "*Hypothesised mechanism*." This will clearly separate Section 3.2 from the observed results and allow it to focus exclusively on the proposed hypothesised interpretation, thereby reinforcing the distinction between empirical findings and mechanistic speculation. See also our response to your comment on result interpretation.
- **Clarifying the narrative:** To further emphasize the putative nature of the mechanism, we propose to adapt its description in various places —e.g. "As the cover develops, *we hypothesise that* the thin cover modifies soil water fluxes" in line 322, "The higher content under the thin cover crop *would* therefore reflect a greater retention" in line 324, "suggesting that it *would* depend not only on the stage of development of the cover" in line 328, etc.— and to insert the following sentence at the end of line 331: "*As evapotranspiration, leaching, microbial activity and metabolites were not analysed, we cannot confirm this hypothesised mechanism.*"

- **Updating the figure caption:** We propose to revise the caption of Figure 4 to read: *"Hypothesised mechanism: cover crops reduce pesticide leaching by modifying soil water fluxes through evapotranspiration, thereby concentrating pesticides in the rhizosphere, where they may be more effectively degraded by soil microbiota."*

These modifications lead to small revisions in lines 366–368, in the caption of Figure 5 and in lines 493 and 544.

In addition, we propose to remove lines 422–423.

**viii)   Water management and potential runoff**

The manuscript does not describe whether the pots were equipped to prevent water runoff or collect leachate, which is particularly relevant given the study's focus on pesticide fate. While soil moisture was monitored and controlled through watering regimes, it is unclear whether pots had drainage holes or whether leached water was collected or measured. If drainage occurred and was not accounted for, this could confound interpretations of pesticide disappearance as degradation versus loss. Please clarify how water balance and potential runoff or leaching were managed throughout the experiment.

Thank you for this important observation. We acknowledge that this detail was missing from the preprint. To address your comment, we propose to add the following clarification at the end of line 131: "*To prevent water runoff and uncontrolled leaching, each pot was placed in an individual saucer with sufficient capacity to retain any excess irrigation water. Saucers were monitored after each watering throughout the experiment, and no overflow was observed, confirming that all drainage water was fully retained*."

We agree that this design, combined with the limited depth of the pots, limits the extrapolation of our results to field conditions, where vertical drainage towards deeper soil horizons occurs regularly (as discussed in lines 495–503). To explicitly acknowledge this limitation, we also propose to add the following sentence at the end of line 519: "*, and sample soil and soil solution at different depths to better assess the vertical mobility of pesticide residues;*"

**c) Technical corrections**

- Line 295: Replace "witch" with "which."
  Thank you for your careful review. This typographical error has been corrected as suggested.
- Lines 73 and 79: Citations to Pelletier and Agnan are missing the publication year.
  The missing publication years have been added for Alletto et al. (line 68) and for Pelletier and Agnan (lines 73 and 79), as recommended.
- Line 80: The statement about bioremediation prioritizing reactors over field applications needs citation. Consider rephrasing or supporting with evidence, as it oversimplifies the field's current practices.
  We propose to rephrase lines 80–84 as follows: "*Despite progress in the literature, two main limitations remain: (1) field research is often limited to a narrow range of pesticide molecules and cover crops, with inconsistent assessments of soil compartments; and (2) influence of cover crops is generally not analysed in relation to the physicochemical properties of the molecules. These gaps prevent a broader understanding of the general*

*applicability of cover crop remediation strategies for different pesticide molecules."* and to remove line 87 in consequence.

**d) Conclusion**

We acknowledge the limitations of our study and hope that the preliminary results presented will support future research efforts and funding aimed at expanding the analytical scope. Ideally, such efforts would include the quantification of a wider range of compounds and their metabolites, the monitoring of plant uptake and microbial dynamics, and the use of an experimental design that accounts for both leaching and evapotranspiration processes.

We trust that the proposed revisions will help to clarify the exploratory nature of the hypothesised mechanism, while highlighting its potential as a basis for future investigations.

**References**

Blanco-Canqui, H., Ruis, S. J., Koehler-Cole, K., Elmore, R. W., Francis, C. A., Shapiro, C. A., Proctor, C. A., & Ferguson, R. B. (2023). Cover crops and soil health in rainfed and irrigated corn: What did we learn after 8 years? *Soil Science Society of America Journal*, *87*(5), 1174–1190. https://doi.org/10.1002/saj2.20566

Oksanen, J., Simpson, G. L., Blanchet, F. G., Kindt, R., Legendre, P., Minchin, P. R., O'Hara, R. B., Solymos, P., Stevens, M. H. H., Szoecs, E., Wagner, H., Barbour, M., Bedward, M., Bolker, B., Borcard, D., Carvalho, G., Chirico, M., Caceres, M. D., Durand, S., … Borman, T. (2025). *vegan: Community Ecology Package* (Version 2.6-10) [Dataset]. https://cran.r-project.org/package=vegan

Wang, Y., Liu, L., Tian, Y., Wu, X., Yang, J., Luo, Y., Li, H., Awasthi, M. K., & Zhao, Z. (2020). Temporal and spatial variation of soil microorganisms and nutrient under white clover cover. *Soil and Tillage Research*, *202*, 104666. https://doi.org/10.1016/j.still.2020.104666

**2) Response to RC2 (anonymous)**

Vandevoorde et al. monitor pesticide residue levels in soil and soil solution under two different cover crop densities. The results showed that, compared to bare soil, thin cover crops reduce pesticide leaching after sowing for 80 days. In addition, well-developed cover crops reduce soil pesticide contents by more than 33%. The experimental design is clear, and the results help to reduce pesticides in soil. I have several suggestions that may improve the MS quality.

**a) Comment 1**

The novelty of the manuscript is not clearly established; for example, including more pesticides in the study should not be regarded as an innovation.

We appreciate this comment and agree that simply increasing the number of pesticides tested is not in itself a novel contribution. The core novelty of our study lies in addressing a specific gap in the literature: while most previous research has focused on the effects of *established* cover crops or mulches on *newly applied* pesticides (see lines 56–58), our study investigates how *newly sown*

*living* cover crops influence the fate of *pre-existing* pesticide residues. This distinction is critical for evaluating the phytoremediation potential of cover crops within active cropping systems (without taking land out of production) and for distinguishing our focus from the more common evaluation of the effects of cover crops on freshly applied pesticides.

The inclusion of multiple pesticide molecules in our study is not an innovation, but a methodological necessity to assess whether the fate of pesticides under cover crops could be predicted based on the physicochemical properties of the active substances and the density of the cover crop. This required a wide range in the values of the physicochemical properties, which required a wide range of selected active substances.

To make this contribution explicit in the manuscript, we propose to revise lines 80–84 on the literature gap as follows: "*Despite progress in the literature, two main limitations remain: (1) field research is often limited to a narrow range of pesticide molecules and cover crops, with inconsistent assessments of soil compartments; and (2) influence of cover crops is generally not analysed in relation to the physicochemical properties of the molecules. These gaps prevent a broader understanding of the general applicability of cover crop remediation strategies for different pesticide molecules.*"

**b) Comment 2**

The key issue is the lack of explanations for the results, e.g., microbial degradation is not involved in the study.

We fully agree that mechanistic processes such as microbial degradation were not directly measured in our study, and we thank the reviewer for pointing out the need to clearly distinguish between observed data and proposed interpretations.

To reinforce this distinction, we propose the following revisions:
- **Restructuring Section 3.2**: We propose to move the results and discussion regarding the behaviour of the three contrasted molecules (lines 312–321, 339–347, and 348–357, as well as Figure 3) to the Supplement and to retitle the section "*Hypothesised mechanism*." This will clearly separate Section 3.2 from the observed results and allow it to focus exclusively on the proposed interpretation, thereby reinforcing the distinction between empirical findings and mechanistic speculation. See also our response to your comment on result interpretation.
- **Clarifying the narrative:** To further emphasize the putative nature of the mechanism, we propose to adapt its description in various places and to insert the following sentence at the end of line 331: "*As evapotranspiration, leaching, microbial activity and metabolites were not analysed, we cannot confirm this hypothesised mechanism.*"
- **Updating the figure 4 caption:** We propose to revise the caption of Figure 4 to read: "*Hypothesised mechanism: cover crops reduce pesticide leaching by modifying soil water fluxes through evapotranspiration, thereby concentrating pesticides in the rhizosphere, where they may be more effectively degraded by soil microbiota.*"
- **General adaptations throughout the manuscript**: We will also refine the wording in several locations to make a clearer distinction between observed results and proposed mechanisms.

**c) Comment 3**

Please carefully check the overall writing of the MS, making it more concise, rigorous, and reliable. e.g., lines 530-535 are not conclusions.

We have reviewed the manuscript with the aim of improving the conciseness, rigour, and precision of the narrative. We propose several cuts and refinements throughout the manuscript to streamline the text and focus on key messages —for example, in lines 30–34, 81–82, 87, 312–321, 339–347, 349–357, 423, 437–445, 517–518 and 530–533, as well as the removal of Figure 3. Specifically, we propose to move lines 312–321, 339–347, and 348–357, along with Figure 3, to the Supplement (see our response to RC1 for details).

In response to your specific point regarding lines 530–533, we agree and propose transfer them to the discussion in line 453.

In order to maintain rigor and reliability, we have retained the Materials and Methods and the core of the Results and Discussion sections intact. Additional improvements to the interpretation and discussion —particularly in distinguishing data from hypotheses— are addressed in our response to you second comment above.

---

## Referee Report (RR1)

**Revision report**

Egusphere-2025-943 - Living cover crops reduce pesticide residues in agricultural soil

The article entitled "Living cover crops reduce pesticide residues in agricultural soil" presents a study conducted in a greenhouse with the overall objectives of evaluating the effects of cover crops on pesticide dynamics in soil and soil solution and relates that with the physicochemical properties of the tested pesticides.

The subject of the article is original and relevant, and the findings are interesting. In particular, I appreciated the study of the pesticide levels not only in soil but also in the soil solution, and the analysis of the results under the light of the physicochemical properties of the active substances that were used. Both of these topics are important to shed light on the possible fate of pesticides in soil and may be of help to interpret the results.

My main concerns are generally focused on some aspects of the methodology and on the conclusions that were derived from the results which, in some cases and in my opinion, are not supported by them. All of these aspects, and some others, are detailed below. The line numbers refer to the manuscript without tracked changes.

Title

The title should emphasize what, in my opinion, are 2 strong points related with this article: the analysis not only of soil, but also of the soil solution and the inclusion of the physicochemical properties of the active substances. In addition to this, the title does not exactly correspond to what was found. While, in general, cover crops were associated with lower pesticide concentrations in soil, in some cases this did not occur, especially for the "thin cover" modality. For example, at day 80 the tebuconazole content in that modality was significantly higher than in the control and the contents of fluxapyroxad, MCPB and pyraclostrobin did not significantly differ between the 2 modalities.

Abstract

A brief synthesis of the methodology should be included in the abstract.

Introduction

L88-89 – As the authors did not evaluate plant uptake of pesticides, they could not test this hypothesis. Therefore, in my opinion, "hypothesised" should be replaced by "considered".

Materials and Methods

Figure 1 – The authors refer that day 0 corresponds to 5 Jan 2024. Considering this, day 45 should correspond to 19 Feb 2024 and day 80 to 25 Mar 2024. However, in Supplementary Material (SM) table S2, the results from 3 sampling days are shown: 5 Jan 2024, 19 Feb 2024, and 29 Apr 2024. Therefore, it appears to exist a discrepancy in the 3rd sampling day between Materials and Methods and SM. Could you please clarify this?

The legend should contain more information. For example, a brief description of the modalities.

L106-107 – Considering that the authors have the information regarding the pesticide applications, including the day of application, the active substances that were applied and their application rates, why not determining the predicted concentrations at the sapling days and compare these values with the measured concentrations?

L115-117 – There are 2 factors varying in the modalities: plant species and plant density/plant biomass. This should be taken into consideration while interpreting the results.

L144-145 and L148-149 – Why were the temperatures of storage different between soil samples and soil solution samples?

L181-186 – I found the list of physicochemical properties that was chosen by the authors comprehensive and well justified.

L204-206 – Why not use the Analysis of Variance (ANOVA) followed by a post-hoc (after checking their assumptions) to compare all the 3 modalities? The 2 t-tests only allowed the comparison between each one of the 2 cover crop modalities and control and not between the 2 cover crop modalities themselves. Of course, a third t-test could be added, but I believe that multiple t-tests increase the probability of Type I errors.

Results and Discussion

L210 – I believe that the application rates refer to what was presented in Table 1. Therefore, that table should be mentioned here. Furthermore, what the authors call "application rates" here is designated by "quantity" in Table 1. This terminology should be homogenised.

L212-213 – In here only iodosulfuron-methyl-sodium and mefenpyr-diethyl are referred, while below (L214) a third active substance is referred as not quantified in all the samples.

L220-254 – I appreciated the effort made by the authors in discussing the obtained results under the light of the pesticide physicochemical properties and I think that this is one of the strong points of the paper. However, I think that the results that are being explored are difficult to be followed by the reader. I noticed that plots were presented in Supplementary Materials, but those plots should be mentioned here. Considering the importance of these results for the discussion, in my opinion, they should be presented here and not in the Supplementary Material. This could be done by either presenting them in Table(s) or in Figure(s).

L251 – I agree that the referred procedure could have induced a bias. However, that was not confirmed by the authors. Therefore, in my opinion, it should be written "This may have introduced a bias…", instead of "This introduced a bias…".

L308-309 – Instead of "highly applied", I believe a more accurate expression would be, for example, "applied at higher application rates".

L387 and L389 – $3.5 \times 10^{-9}$ and $1.3 \times 10^{-4}$ (the "×" symbol is missing).

L390 – "suggesting" instead of "suggestion"

L409-410 – I suggest using the format (author(s), year) for the referred sources.

L417 – Highly volatile compounds are primarily lost to the atmosphere and not to ground water. So, solubility should be the most important aspect here.

L450-451 – As other effects are possible, it is not certain "that any practice that increases living cover crop and microbial activity will contribute to pesticide degradation." Consider replacing "will contribute" by "may contribute" or something equivalent. In addition to this, I believe that instead of "crop cover" the authors meant "cover crop".

L464-465 – In my opinion, the authors cannot separate the effects of cover density from the effects of the species that were tested. For that, it would be necessary to test the same density with different species and the same species with different densities.

L472-473 – Considering what is shown by Figure S1 from Supplement S4, while the results show that the content of mefentrifluconazole in the modality "thick cover" was lower than in the control (bare soil), its content in the modality "thin cover" was actually higher. Therefore, in my opinion, the accuracy of this statement should be improved.

Conclusions

L516 – The results obtained by the authors show that in many cases the cover crop modalities were associated with lower pesticide contents. However, in some cases no significant differences were found, and in others, a higher pesticide content was found. Therefore, I would suggest increasing the accuracy of the sentence.

L537-538 – While the reduction in pesticide use, especially the most hazardous ones, is a desirable goal, it is not backed-up by the findings presented in this paper. Therefore, the accuracy of this sentence should be improved.

Supplementary Material

The legends of the Tables and Figures should contain more information. For example, when applicable the details regarding the different modalities should be described and the units of measurement should be indicated.

If I am not mistaken most of the Supplementary Materials are not referred in the main manuscript. Please confirm this and correct it when applicable.

Tables S3 and S4 – The meaning of "LQ" should be explicit.

Table S4 – The meaning of "ND" is not explained.

Table S4 – The authors referred that a total of 18 active substances were analysed in the samples. However, only the results from 14 active substances are presented in this table.

Supplements S4 and S6 – The results refer not only to the pesticide contents in soil, but also to the pesticide contents in the soil solution. However, the y axis refers only to soil (i.e., the unit is µg kg$^{-1}$ of fresh soil). Please correct this.

---

## Referee Report (RR2)

The article by Vandervoorde et al. investigates how the presence of living plant cover, at different densities, influences the degradation of a pesticide mixture in soil. This topic is particularly relevant for understanding the environmental fate of pesticides and for advancing sustainable agricultural practices. The authors analyzed the degradation of 18 commonly used pesticides with diverse physicochemical properties under two different crop cover conditions. They monitored pesticide concentrations in both soil and soil solution and proposed a quantification of degradation in relation to pesticide properties.

Although the experimental setup - which included ten replicates and covered a wide range of pesticides under greenhouse pot conditions - provides valuable insights into pesticide behavior, the conclusions drawn in the paper appear to lack robustness. Specifically, the authors suggest that differences in residual pesticide concentrations result from variations in crop evapotranspiration and microbial degradation near the rhizosphere; however, no direct measurements were made in plant tissues, evaporated water (despite the greenhouse setup), or pesticide metabolites. Furthermore, the introduction mentions possible interactions among pesticides affecting their mobility (lines 33–34), which may complicate the interpretation of individual pesticide behavior when applied as a mixture. Nevertheless, despite these limitations, the study effectively highlights contrasting trends in pesticide mobility depending on land cover, and clearly relates them to well-known pesticide properties such as water solubility, molecular weight, and vapor pressure.

Other comments:
Introduction: The introduction mentioned "pesticides" as a whole while the results focused on differences in physico-chemical properties. The introduction (which is relatively long) could develop on these properties and then better explain the novelty of the results vs expected behavior knowing pesticides characteristic's.
l. 28: "… diffuse contamination of other environmental compartments" I would like to have a rough quantification of this dispersion

l. 31-34 : "chlordecone adsorbed on soil particles is currently being transported to surface and groundwater bodies by soil erosion (enhanced by bare soils resulting from contemporary glyphosate applications)" not clear, please rephrase and shorten the whole sentence.

l.44 – 47 :" chlordecone adsorbed on soil particles is currently being transported to surface and groundwater bodies by soil erosion (enhanced by bare soils resulting from contemporary glyphosate applications) " please argument what chlordecone degradation (cf l.29-30) is limited

l. 59-60 " "enhancing microbial activity…" isn't that the definition of phytoremediation given l.43-45.

L 61 "the mineralisation of 2,4-D." what is 2,4 -D ???

l.62-65 sentence too long

l.69 "They highlighted the importance of soil organic carbon" be consistent during all manuscript between organic carbon and organic matter

l. 68-71 "They highlighted the importance of soil organic carbon and cover biomass production in reducing leaching, with cover crops producing over 2 tDM ha−1 significantly reducing leaching in contrast to no effects observed at 0.3 tDM ha−1 (DM: dry matter)." Not very clear, to what 0.3tDM refers, bare soil? why not null in this case?

Methods:
l.129 Was temperature maintained at 20 even during night?

l.155 "No metabolites were quantified" not sure if they were not found or not searched

l.164 – 175: The paragraph is not very clear (but it is a good point to explain)

l.176: split the paragraph in two parts (pesticide properties and data treatment)

l.178-182 the importance of different properties should have been introduced before.

l.184 "data pre-analyses were performed in MS excel" what are pre-analyses?

l.185: put the sentence about R version (Rstudio doesn't matter) at the end of the paragraph.

l. 200 : Which package/ function are used for the deviation tests?

Results:
l. 243-245: I don't understand the explanation about reduced soil mass, please rephrase.
l. 290: extra point in "…60% of the variance. Separated.."

l. 290-300 : "The first dimension, accounting for 60 % of the variance, separated the molecules in two groups: (1) negative values corresponded to substances such as mefentrifluconazole and tebuconazole, which have high soil sorption, high lipophilicity, low water solubility and/or long soil persistence; and (2) positive values corresponded to substances such as clopyralid or pyroxsulam, which have low soil sorption, low lipophilicity, high water solubility and/or short soil persistence. "**1.** It is not entirely clear which data were included in the PCA analysis. Did the authors use only the percentage of the initial pesticide concentration at each sampling date, or were physicochemical properties and sampling compartments also incorporated? In line 302, the statement *"reflecting a shift towards a dominance of molecules with higher soil sorption, bioconcentration or persistence"* is ambiguous, as it is unclear which part of this interpretation is directly supported by the PCA results and which derives from the known properties of the molecules.

l. 302 please define "post-emergence"

Fig. 2: I think that it can be useful to have subpanels for left and right. Also, I don't understand what is the right plot.

l. 320 -340: The interpretations and proposed mechanisms, particularly those related to water fluxes and the effects of crop density, are insufficiently supported by evidence..

l. 482: "Our main hypothesis highlighted the role of microorganisms in pesticide biodegradation, but we were unable to directly monitor microbial activity ". In my opinion, this sentence-and possibly the introduction as well - should be reformulated to clearly emphasize the main hypothesis and how the study was designed to address it. The statement *"our study was not designed to test our hypothesis"* seems inappropriate, as it undermines the scientific rationale and clarity of the research objective

---

## Author Response (AR2)

egusphere-2025-943

**Living cover crops reduce pesticide residues in agricultural soil**

Author's response to the referees

Dear Referees and Editors,

Thank you for your time and for the thoughtful and constructive review you provided. We believe that your comments and suggestions offer valuable insights that have helped us improve the overall quality of the manuscript.

We would like to note that the comments provided by (anonymous) referee #4 appear to be based on the initial submitted version of the manuscript as this is the only version for which the line numbering corresponds to the referee's comments. Because substantial revisions were made during the first revision round (including removal or restructuring of entire sections and the relocation of some material to the Supplements) several of referee's #4 comments no longer correspond to the current content of the manuscript. We have addressed each of their comment wherever possible; however, in a few instances the referenced text no longer exists in its original form, or the issue has already been resolved in previous revisions.

As mentioned in our communication with the editors in August, we had submitted an updated third version of the manuscript based on feedback receive following a presentation of the results and our PhD defence. Unfortunately, this revised version was not considered during the review process and was therefore not the one assessed by the referees. As a result, the changes tracked in the revised manuscript discussed here incorporates both our responses to the comments of referees #3 and #4, and the revisions previously submitted in August to the editors.

Please find below our detailed responses to the referees' comments. For clarity, we have reproduced their original comments in **black**, provided our responses in **blue** and highlighted major proposed changes in **green**. All line references correspond to the newly revised manuscript **with tracked changes**.

**Table of contents**

**1) Responses to Referee #3 (Abel Veloso)**

The article entitled "Living cover crops reduce pesticide residues in agricultural soil" presents a study conducted in a greenhouse with the overall objectives of evaluating the effects of cover crops on pesticide dynamics in soil and soil solution and relates that with the physicochemical properties of the tested pesticides.

The subject of the article is original and relevant, and the findings are interesting. In particular, I appreciated the study of the pesticide levels not only in soil but also in the soil solution, and the analysis of the results under the light of the physicochemical properties of the active substances that were used. Both of these topics are important to shed light on the possible fate of pesticides in soil and may be of help to interpret the results.

My main concerns are generally focused on some aspects of the methodology and on the conclusions that were derived from the results which, in some cases and in my opinion, are not supported by them. All of these aspects, and some others, are detailed below. The line numbers refer to the manuscript without tracked changes.

**a) Title**

The title should emphasize what, in my opinion, are 2 strong points related with this article: the analysis not only of soil, but also of the soil solution and the inclusion of the physicochemical properties of the active substances. In addition to this, the title does not exactly correspond to what was found. While, in general, cover crops were associated with lower pesticide concentrations in soil, in some cases this did not occur, especially for the "thin cover" modality. For example, at day 80 the tebuconazole content in that modality was significantly higher than in the control and the contents of fluxapyroxad, MCPB and pyraclostrobin did not significantly differ between the 2 modalities.

> Thank you for this very important comment. Although we would argue soil solution is contained within soil and that mentioning it in the title would make it heavy, we agree with the rest of your comment and propose the following update of the title, with the inclusion of a subtitle:
>
> > Living cover crops alter the fate of pesticide residues in soil: influence of pesticide physicochemical properties.

**b) Abstract**

(Lines 7–11) A brief synthesis of the methodology should be included in the abstract.

> We propose revising the abstract to include more insight of the method (lines 7–11):
>
> > The objective of this study was to evaluate to what extent pesticide residues with contrasting physicochemical properties are affected by living cover crops. We conducted a greenhouse experiment testing two cover crop densities against a bare soil control, and quantified residues (by LC-QTOFMS) of 18 pesticide ingredients (active substances and safeners) in both soil and soil solution. We then

related the observed reduction in residues to key physicochemical properties of the pesticide ingredients.

**c) Introduction**

L88-89 (line 104) – As the authors did not evaluate plant uptake of pesticides, they could not test this hypothesis. Therefore, in my opinion, "hypothesised" should be replaced by "considered".

Thank you for this observation. We corrected the manuscript as suggested.

**d) Materials and Methods**

Figure 1 – The authors refer that day 0 corresponds to 5 Jan 2024. Considering this, day 45 should correspond to 19 Feb 2024 and day 80 to 25 Mar 2024. However, in Supplementary Material (SM) table S2, the results from 3 sampling days are shown: 5 Jan 2024, 19 Feb 2024, and 29 Apr 2024. Therefore, it appears to exist a discrepancy in the 3rd sampling day between Materials and Methods and SM. Could you please clarify this?

Thank you for your careful reading of both the main text and the supplements. The date indicated in Table S2 of the Supplements S1 is indeed incorrect. It corresponds to the date on which we received the analytical results from our co-author's laboratory, rather than the sampling date. We have corrected this in the revised version of the Supplements.

The legend should contain more information. For example, a brief description of the modalities.

The caption was modified to better describe the experimental method, with the following inclusion:

Homogenised organic soil was potted on day –18 and treated with 18 pesticide ingredients on day –14, then sown on day 0 with two cover types (a *thick* winter spelt and a *thin* multi-species mix) or left bare (*n*=35 pots total). Greenhouse growth was monitored and soil, soil solution and plant biomass were sampled on days 0, 45 and 80.

L106-107 – Considering that the authors have the information regarding the pesticide applications, including the day of application, the active substances that were applied and their application rates, why not determining the predicted concentrations at the sapling days and compare these values with the measured concentrations?

Thank you for this suggestion. While we agree that modelled concentrations could, in principle, be compared with the measured values, this was not the objective of the present study. Our aim was to assess how different cover-crop densities influence pesticide dynamics relative to a common baseline (the bare-soil control), rather than to evaluate model performance or compare observations to predicted degradation curves. In addition, implementing such comparisons would require selecting and justifying a specific degradation model and discussing its assumptions and limitations. Given the diversity of active substances applied (with contrasting physicochemical properties, environmental behaviours and degradation pathways), using simple first-order degradation based solely on DT50 values would be overly simplistic and potentially

misleading. A more sophisticated modelling approach would require substantial additional work and fall outside the scope of this study.

For these reasons, we opted to use the bare soil (control) as our reference, which we consider both more robust for our objectives and more directly interpretable.

L115-117 (lines 145-157) – There are 2 factors varying in the modalities: plant species and plant density/plant biomass. This should be taken into consideration while interpreting the results.

Thank you for highlighting this important point. We fully agree that both plant species and plant biomass differed between the two cover-crop modalities, and that this has implications for interpreting the results. This issue was already discussed in detail in the manuscript version we submitted in August to the Editors, but which was unfortunately not forwarded to the reviewers. The current revised manuscript explicitly addresses this at two stages: in the description of the experimental design (lines 145-157) and in the Discussion (lines 530–539).

Lines 145-157 — Three cover modalities were tested (Fig. 1). Two types of cover crops with rapid growth: (1) ten pots with winter spelt (*Triticum spelta*) and (2) ten pots with a multi-species cover (20 % buckwheat, *Fagopyrum esculentum*; 20 % phacelia, *Phacelia tanacetifolia*; 20 % vetch, *Vicia villosa*; and 40 % white mustard, *Sinapis alba*; seed w/w); in addition to 15 pots kept bare as a control (for a total of 35 pots in the experiment). In the following, we refer to the cover crops as *cover types*, while cover types together with the control are collectively referred to as *cover modalities*. The two cover types were sown on 5 January 2024 (day 0) at a density of $191 \pm_{sd} 12 \pm_{\Delta} 1$ $kg_{seeds}$ $ha^{-1}$ (winter spelt; $n$=10) and $147 \pm_{sd} 3 \pm_{\Delta} 1$ $kg_{seeds}$ $ha^{-1}$ (multi-species mix; $n$=10), respectively, with the expectation of similar shoot biomass. However, they reached a shoot biomass of $0.43 \pm_{sd} 0.04 \pm_{\Delta} 0.07$ $t_{DM}$ $ha^{-1}$ and $0.25 \pm_{sd} 0.08 \pm_{\Delta} 0.04$ $t_{DM}$ $ha^{-1}$, respectively, on day 45 ($n$=5), and a shoot biomass of $1.12 \pm_{sd} 0.02 \pm_{\Delta} 0.18$ $t_{DM}$ $ha^{-1}$ and $0.36 \pm_{sd} 0.09 \pm_{\Delta} 0.06$ $t_{DM}$ $ha^{-1}$, respectively, on day 80 ($n$=5). This difference in biomass production may be due to the phytotoxic effect of the applied pesticides to the multi-species mix. Consequently, we analysed pesticide content in relation to biomass difference (referred to as *cover density*) rather than species difference between the covers, comparing the *thick* winter spelt cover and the *thin* multi-species cover mix with the bare control.

Lines 530–539 — Building on this limitation, our analysis focused on above-ground biomass density as the primary indicator, despite the cover crops comprising different species. This approach was motivated by the markedly different development patterns of the two cover types. Interestingly, at comparable biomass densities (day 45 for the thick cover and day 80 for the thin cover), pesticide behaviour appeared similar. This suggests that shoot biomass density — used here as a proxy for root development— may be more influential than species composition in determining pesticide dynamics. Therefore, selecting cover crop species that can tolerate residual pesticides and establish rapidly may have a greater impact on mitigating pesticide transfer than maximising species diversity. While this prevents a direct evaluation of species-specific effects, it highlights the importance of biomass development. Furthermore, the poor establishment of the thin cover crop may have resulted from the phytotoxic effects of the applied

pesticides. This hypothesis warrants further investigation, including the use of control pots growing cover crops without pesticide residues.

L144-145 and L148-149 – Why were the temperatures of storage different between soil samples and soil solution samples?

Soil and soil solution samples were stored under different conditions due to their distinct stability characteristics. Soil samples can be safely frozen without affecting pesticide concentrations, which allowed us to accumulate samples and perform extractions and analyses simultaneously, thereby reducing inter-sample variability. In contrast, freezing soil solution can induce degradation of pesticides, leading to lower measured concentrations upon thawing. Therefore, soil solution samples were stored at 4 °C and analysed within seven days of collection to preserve their integrity.

L181-186 – I found the list of physicochemical properties that was chosen by the authors comprehensive and well justified.

Thank you for this feedback.

L204-206 – Why not use the Analysis of Variance (ANOVA) followed by a post-hoc (after checking their assumptions) to compare all the 3 modalities? The 2 t-tests only allowed the comparison between each one of the 2 cover crop modalities and control and not between the 2 cover crop modalities themselves. Of course, a third t-test could be added, but I believe that multiple t-tests increase the probability of Type I errors.

Thank you for this important comment. We agree that an ANOVA followed by a post-hoc test (e.g., Tukey) would normally provide a more rigorous framework for comparing all three modalities, including the two cover-crop treatments. However, as you noted yourself in an earlier comment, the two cover-crop modalities differ not only in density but also in species composition. These differences make a direct statistical comparison between the two cover types difficult to interpret and lead to potentially misleading conclusions. For this reason, we chose to limit the analysis to comparisons between each cover-crop type and the control. We therefore did not pursue a full three-way comparison.

We propose to clarify that in the Material and Method (lines 243–248) as follows:

To assess whether the differences in pesticide content were statistically significant, we performed individual unilateral *t*-tests for each cover-crop type versus the control (implemented in MS Excel using the `T.DIST.RT` function). We limited the analysis to pairwise comparisons with the control because the two cover-crop types differ not only in density but also in species composition, making direct statistical comparisons between them difficult to interpret. These tests therefore evaluate whether the concentration difference between each cover type and the control is significantly different from zero (positive or negative).

**e) Results and Discussion**

L210 (line 255) – I believe that the application rates refer to what was presented in Table 1. Therefore, that table should be mentioned here. Furthermore, what the authors call "application rates" here is designated by "quantity" in Table 1. This terminology should be homogenised.

> Thank you for this careful observation. As suggested, Table 1 is now explicitly referenced at line 255. In addition, the terminology has been standardised throughout the manuscript: we now consistently refer to these values as the "applied dose" (d) to avoid ambiguity.

L212-213 (lines 256–260) – In here only iodosulfuron-methyl-sodium and mefenpyr-diethyl are referred, while below (L214) a third active substance is referred as not quantified in all the samples.

> The two statements refer to different situations: iodosulfuron-methyl-sodium and mefenpyr-diethyl showed no detection, meaning they were not detected in *any* sample. In contrast, pinoxaden was detected *in some* samples but not in all.

L220-254 – I appreciated the effort made by the authors in discussing the obtained results under the light of the pesticide physicochemical properties and I think that this is one of the strong points of the paper. However, I think that the results that are being explored are difficult to be followed by the reader. I noticed that plots were presented in Supplementary Materials, but those plots should be mentioned here. Considering the importance of these results for the discussion, in my opinion, they should be presented here and not in the Supplementary Material. This could be done by either presenting them in Table(s) or in Figure(s).

> Thank you for highlighting the importance of the results on pesticide behaviour by physicochemical properties. While we agree that these results are key, we believe that the main focus should remain on the subsequent sections, and adding additional tables or figures to the first section of the results would make it overly dense. Moreover, the manuscript has been criticized for length, with a recommendation to include material in the Supplements. Therefore, to ensure transparency and guide the reader, we have now explicitly referred in line 229 to the relevant Supplements where raw data and additional Figures are provided.

L251 (line 298) – I agree that the referred procedure could have induced a bias. However, that was not confirmed by the authors. Therefore, in my opinion, it should be written "This may have introduced a bias…", instead of "This introduced a bias…".

> Thank you for this comment. Line 298 has been revised as suggested.

L308-309 (lines 359) – Instead of "highly applied", I believe a more accurate expression would be, for example, "applied at higher application rates".

> Thank you for this suggestion. Line 359 has been revised as suggested.

L387 and L389 – 3.5 × 10-9 and 1.3 × 10-4 (the "×" symbol is missing).

Thank you for noticing this. The "×" symbol has been corrected throughout the manuscript.

L390 (line 446) – "suggesting" instead of "suggestion"

Thank you for this careful proofread. Line 446 has been revised as suggested.

L409-410 (lines 462–466) – I suggest using the format (author(s), year) for the referred sources.

Thank you for the suggestion regarding citation format. We have revised the sentence to adopt the author–year style while explicitly indicating the version of the databases used. In particular, the PPDB database is cited as Lewis et al. (2016), with the version accessed in May 2024, and Phytoweb data are noted as extracted in November 2024. This ensures transparency and reproducibility, while making clear that the results are based on compiled datasets:

In Wallonia (southern half of Belgium), 141 authorised active substances — including 30 % of the most frequently used active substances in the period 2015–2020 (Corder, 2023)— fall within all three thresholds and mainly concern potato, sugar beet and winter cereal crops (Lewis et al., 2016, version accessed May 2024; *phytoweb.be*, data extracted November 2024).

L417 (lines 474–476) – Highly volatile compounds are primarily lost to the atmosphere and not to ground water. So, solubility should be the most important aspect here.

Thank you for highlighting the potential for misreading. What we meant is:

Although this effect may be limited for highly volatile pesticides (which are lost to the atmosphere before cover crops can affect them) and for soluble molecules (which may leach before cover crops establish), it represents an important step in phytoremediation.

L450-451 (line 511) – As other effects are possible, it is not certain "that any practice that increases living cover crop and microbial activity will contribute to pesticide degradation." Consider replacing "will contribute" by "may contribute" or something equivalent. In addition to this, I believe that instead of "crop cover" the authors meant "cover crop".

Thank you for this comment. Line 511 has been revised as suggested.

L464-465 (line 530–539) – In my opinion, the authors cannot separate the effects of cover density from the effects of the species that were tested. For that, it would be necessary to test the same density with different species and the same species with different densities.

Thank you for this important comment. We fully agree that, in our experimental setup, species composition and biomass density varied simultaneously, and that our design does not allow these two factors to be fully disentangled. As noted in our detailed response above, this limitation is acknowledged in the revised manuscript in both the experimental design (lines 145–157) and the discussion (lines 530–539).

L472-473 (line 545) – Considering what is shown by Figure S1 from Supplement S4, while the results show that the content of mefentrifluconazole in the modality "thick cover" was lower than in the control (bare soil), its content in the modality "thin cover" was actually higher. Therefore, in my opinion, the accuracy of this statement should be improved.

> Thank you for this careful observation. To avoid overstating the result, we have revised the sentence to explicitly specify that the reduction applies to the thick cover only.

**f) Conclusions**

L516 (lines 593–597) – The results obtained by the authors show that in many cases the cover crop modalities were associated with lower pesticide contents. However, in some cases no significant differences were found, and in others, a higher pesticide content was found. Therefore, I would suggest increasing the accuracy of the sentence.

> Thank you for this valuable comment. We have revised the conclusion to increase its accuracy and ensure it reflects the full range of observed outcomes, including cases with no significant differences or higher pesticide contents under certain cover-crop modalities. We believe the updated wording aligns with the new title and more explicitly distinguishes between retention and degradation processes. The revised conclusion (lines 593–597) now reads:

> > Our results show that living cover crops alter the fate of pesticide residues in soil through two complementary mechanisms: retention of residues in the topsoil under low biomass, and enhanced degradation under higher biomass, both influenced by the physicochemical properties of the pesticides. These mechanisms limit pesticide movement beyond the soil profile, highlighting the potential of cover crops to mitigate pesticide transfer to groundwater and other environmental compartments.

L537-538 – While the reduction in pesticide use, especially the most hazardous ones, is a desirable goal, it is not backed-up by the findings presented in this paper. Therefore, the accuracy of this sentence should be improved.

> Thank you for this comment. We have removed the last two sentences to avoid overstatement, ensuring the text reflects only the findings presented.

**g) Supplementary Material**

The legends of the Tables and Figures should contain more information. For example, when applicable the details regarding the different modalities should be described and the units of measurement should be indicated.

> We have updated the figure and table captions accordingly: Tables S3, S4 and S7 have been updated to include units of measurement and all captions now detail the modalities as follows:

> > The *thick cover modality* refers to the winter spelt cover (reaching a shoot biomass of 1.12 $t_{DM}$ ha$^{-1}$ on day 80) and the *thin cover modality* refers to the multi-species mix (reaching a shoot biomass of 0.36 $t_{DM}$ ha$^{-1}$ on day 80).

If I am not mistaken most of the Supplementary Materials are not referred in the main manuscript. Please confirm this and correct it when applicable.

We have carefully checked all references to Supplementary Materials:

— Supplementary Material S1:
- o Table S1 is referred in the main text at line 137–139;
- o Table S2 is referred in the main text at line 164;
- o Tables S3 and S4 are referred in the main text at line 196;
- o Tables S5 and S6 are referred in the main text at line 218–219.
— Supplementary Material S2:
- o Referred in the main text at line 195;
- o Table S7 is indeed not referred in the main text, but is cited within Supplement S2 itself (line 43).
— Supplementary Materials S3 and S4:
- o Referred in the caption of Figures 2 and 4;
- o At lines 370–371, 404–405;
- o At line 546 (Supplement S4 only);
- o Within the Supplementary Material at line 8 (and in the caption of Figures S3;
- o Figure S1 is cited only within Supplement S4 itself (lines 87, 95 and 104).
— Supplementary Material S5:
- o Referred in the main text at line 453 and 456;
- o Table S8 is referred in the main text at line 456 and within Supplement S5 itself (line 123).
— Supplementary Material S6:
- o Referred in the main text at line 406:

Figures S2 and S3 were not explicitly cited in the main text. We have now added references to them at lines 252–253 and 406.

Tables S3 and S4 – The meaning of "LQ" should be explicit.

As suggested, we have defined "LQ" explicitly in the table captions.

Table S4 – The meaning of "ND" is not explained.

As suggested, we redefined "ND" explicitly in the table caption.

Table S4 – The authors referred that a total of 18 active substances were analysed in the samples. However, only the results from 14 active substances are presented in this table.

As stated on line 6 of the Supplementary Material, mefenpyr-diethyl ($LQ_{soil\ solution}$ = 0.15 µg $L^{-1}$) and halauxifen-methyl ($LQ_{soil\ solution}$ = 0.03 µg $L^{-1}$) were never detected in soil solution samples (ND for all samples) and were omitted from Table S4.

Supplements S4 and S6 – The results refer not only to the pesticide contents in soil, but also to the pesticide contents in the soil solution. However, the y axis refers only to soil (i.e., the unit is µg kg-1 of fresh soil). Please correct this.

As explained (lines 211–215) in the main text, in order to allow a direct comparison of the levels of active substances between the two compartments, we have converted the concentrations in soil solution to equivalent fresh soil content (in $\mu g\ kg^{-1}$) by multiplying them by the fraction of soil solution per unit mass of fresh soil, bearing in mind that the soil content also includes some of the soil solution concentration. We repeated this in Supplement S6 to limit ambiguity.

**2) Responses to Referee #4 (anonymous)**

The article by Vandervoorde et al. investigates how the presence of living plant cover, at different densities, influences the degradation of a pesticide mixture in soil. This topic is particularly relevant for understanding the environmental fate of pesticides and for advancing sustainable agricultural practices. The authors analyzed the degradation of 18 commonly used pesticides with diverse physicochemical properties under two different crop cover conditions. They monitored pesticide concentrations in both soil and soil solution and proposed a quantification of degradation in relation to pesticide properties.

Although the experimental setup - which included ten replicates and covered a wide range of pesticides under greenhouse pot conditions - provides valuable insights into pesticide behavior, the conclusions drawn in the paper appear to lack robustness. Specifically, the authors suggest that differences in residual pesticide concentrations result from variations in crop evapotranspiration and microbial degradation near the rhizosphere; however, no direct measurements were made in plant tissues, evaporated water (despite the greenhouse setup), or pesticide metabolites. Furthermore, the introduction mentions possible interactions among pesticides affecting their mobility (lines 33–34), which may complicate the interpretation of individual pesticide behavior when applied as a mixture. Nevertheless, despite these limitations, the study effectively highlights contrasting trends in pesticide mobility depending on land cover, and clearly relates them to well-known pesticide properties such as water solubility, molecular weight, and vapor pressure.

Other comments:

**a) Introduction:**

The introduction mentioned "pesticides" as a whole while the results focused on differences in physico-chemical properties. The introduction (which is relatively long) could develop on these properties and then better explain the novelty of the results vs expected behavior knowing pesticides characteristic's.

> Thank you for this insightful comment. We agree that the introduction should better prepare the reader for the focus on physicochemical properties developed later in the manuscript. In response, we have added explicit references to these properties in the body of the introduction (lines 29–30 and 94) and substantially revised the final two paragraphs to more clearly articulate their relevance, the knowledge gaps identified in the literature, and the specific novelty of our study.
>
> We also note that your observations appear to refer to the initial manuscript submitted. During the first round of review and further revisions sent to the Editors in August, additional modifications were already made to the introduction; we believe the new revisions introduced in response to your comment further strengthen the Introduction. The revised ending of the introduction now reads as follows (lines 94–114):
>
>> To address these gaps, we conducted a controlled, three-month greenhouse experiment designed to evaluate the ability of newly sown cover crops to influence the dynamics of existing pesticide residues in soil and soil solution. Specifically,

we focused on determining whether differences in pesticide behaviour could be related to their physicochemical properties. For this purpose, we monitored the temporal evolution of 18 active substances and two safeners under three modalities: a control (bare soil) and two contrasting living cover crops densities.

Based on the literature, we considered that cover crops may reduce pesticide leaching primarily by modifying soil water fluxes through evapotranspiration, thereby concentrating pesticides near the roots and prolonging their retention within the microbiologically active rhizosphere where bio-degradation is enhanced. Furthermore, following the literature review by Tarla et al. (2020), we considered that rhizosphere-mediated processes play a more important role than plant uptake in controlling pesticide residue dynamics under cover crops. Our main hypothesis was that the influence of cover crops on pesticide dynamics depends on both the physicochemical properties of the molecules and the characteristics of the cover crop. Accordingly, our main objective was to identify trends linking pesticide physicochemical properties with their responses to cover-crop treatments. This included evaluating thresholds in both key molecular properties and cover-crop development that determine whether cover crops exert a measurable effect on residue dynamics in both soil and soil solution compartments. Because our focus was on residue behaviour within soil compartments, rather than on quantifying microbial processes or plant uptake, microbiological monitoring and plant tissue analyses were not included in the study.

l. 28: "… diffuse contamination of other environmental compartments" I would like to have a rough quantification of this dispersion

Thank you for this comment. In the literature, volatilisation, spray drift, runoff, and leaching are recognised as the main processes contributing to pesticide transfer to non-target environmental compartments, with magnitudes varying widely depending on the molecule physicochemical properties, formulation, weather conditions, and cropping system. For volatilisation alone, reported losses range from a few percent to several tens of percent, with rare extreme cases reaching higher values. For example:

— Bedos et al. (2002) reported that volatilisation can reach up to ~90% for certain highly volatile compounds under favourable conditions, although such cases are uncommon.
— Gish et al. (2017) documented more typical losses of 5–25% under field conditions.
— Ferrari et al. (2003) observed volatilisation ranging 5–41% across different pesticides.
— Leistra et al. (2006) reported around 65% volatilisation for chlorpyrifos.
— Loubet et al. (2025) measured 20–50% volatilisation for chlorothalonil.

These studies illustrate the large variability in losses, driven by pesticide physicochemical properties, weather conditions (temperature, wind, humidity), soil moisture, application method, crop cover, etc.

As this is not the main topic of our paper —and in light of your earlier remark encouraging a more concise introduction— we believe that a review of transfer-process quantification would be out of scope and would distract from the main focus of the study. However, we can add a reference to the scientific work of Leenhardt et al. (2023) that discuss this topic (line 34). Therefore, we opt not to expand this section in the manuscript.

l. 31-34 : "chlordecone adsorbed on soil particles is currently being transported to surface and groundwater bodies by soil erosion (enhanced by bare soils resulting from contemporary glyphosate applications)" not clear, please rephrase and shorten the whole sentence.

Thank you for this comment. In a manuscript version we submitted in August to the Editors, but which was unfortunately not forwarded to the reviewers, we removed the example referring to chlordecone as it was indeed unclear and contributed to an unnecessarily long introduction. The revised introduction now presents the context more concisely.

l.44 – 47 :" chlordecone adsorbed on soil particles is currently being transported to surface and groundwater bodies by soil erosion (enhanced by bare soils resulting from contemporary glyphosate applications) " please argument what chlordecone degradation (cf l.29-30) is limited

See our response to your previous comment.

l. 59-60 (line 64) " "enhancing microbial activity…" isn't that the definition of phytoremediation given l.43-45.

Thank you for this careful reading. You are correct that the phrase "enhancing microbial activity" corresponds to the concept of biostimulation introduced earlier. We have revised line 64 accordingly.

L 61 "the mineralisation of 2,4-D." what is 2,4 -D ???

2,4-D (ISO name for 2,4-dichlorophenoxyacetic acid) is one of the oldest and most widely used herbicides and defoliants worldwide, commercially available since 1945 and now produced by many companies following patent expiration. Given its widespread recognition in the field, we consider the current abbreviation sufficient. To answer your comment, we have clarified (line 67) that it refers to a herbicide molecule.

l.62-65 (lines 69–72) sentence too long

Thank you for your comment. We have split the sentence in two, as suggested. It now reads (lines 69–72):

Similarly, multi-year field studies reported reductions in pesticide concentrations under cover crops compared to bare soil. Potter et al. (2007) observed decreases of up to 33 % for atrazine in groundwater under sunn hemp (*Crotalaria juncea*), while White et al. (2009) reported reductions of up to 41 % for metolachlor in soil.

l.69 (lines 78 and 83) "They highlighted the importance of soil organic carbon" be consistent during all manuscript between organic carbon and organic matter

Thank you for your detail review. We have uniformised our use of the term "organic matter" (see lines 78 and 83).

l. 68-71 (lines 77–79) "They highlighted the importance of soil organic carbon and cover biomass production in reducing leaching, with cover crops producing over 2 tDM ha−1 significantly reducing leaching in contrast to no effects observed at 0.3 tDM ha−1 (DM: dry matter)." Not very clear, to what 0.3tDM refers, bare soil? why not null in this case?

Thank you for this comment. To clarify, the sentence refers to cover crops producing 0.3 $t_{DM}$ ha$^{-1}$, not bare soil. At this low biomass, no significant reduction in leaching was observed, whereas cover crops producing over 2 $t_{DM}$ ha$^{-1}$ significantly reduced leaching. The manuscript has been revised accordingly for clarity as follows (lines 77–79):

They highlighted the importance of soil organic matter and cover biomass production in reducing leaching: cover crops producing over 2 $t_{DM}$ ha$^{-1}$ significantly reducing leaching, whereas no significant effect was observed at 0.3 $t_{DM}$ ha$^{-1}$ (DM: dry matter).

**b) Methods:**

l.129 Was temperature maintained at 20 even during night?

Yes, the temperature in the greenhouse was maintained constant, at 20.8 $\pm_{sd}$ 1.6 °C during the whole experiment.

l.155 "No metabolites were quantified" not sure if they were not found or not searched

Thank you for this comment. To clarify, metabolites were not quantified in this study because the laboratory protocol did not allow their analysis. We clarified the revised manuscript to clarify this as follows:

The quantification of metabolites was not pursued due to laboratory protocol limitations.

l.164 – 175 (lines 201–215): The paragraph is not very clear (but it is a good point to explain)

Thank you for this comment. We have simplified the paragraph to improve clarity (see lines 201–215):

The presence of residual moisture in micropores after gravitational drainage means that fresh soil samples contain compounds both adsorbed to soil particles and dissolved in the residual soil solution. For low solubility compounds, the contribution of the residual solution to the measured soil content is minimal. However, for highly soluble, low-volatility substances (e.g. flonicamid, pyroxsulam), the concentration in the residual solution may exceed that adsorbed to soil particles, potentially introducing bias. Drying soil samples prior to analysis does not resolve this issue, as low-volatility compounds remain in the soil while other substances may volatilise during the drying, introducing further bias. This limitation applies broadly to studies quantifying pesticides in soil and complicates comparisons with soil solution measurements. In this study, it prevented us from determining a total mass balance simply by combining soil content and soil solution concentration, as the residual soil solution would effectively be double

counted. Nevertheless, to allow direct comparison between compartments, we converted soil solution concentration to an equivalent fresh soil content (in µg kg$^{-1}$) by multiplying by the fraction of soil solution per unit mass of fresh soil, noting that the soil content inherently includes some of the soil solution.

l.176 (line 216): split the paragraph in two parts (pesticide properties and data treatment)

The section has been split in two parts, as suggested.

l.178-182 the importance of different properties should have been introduced before.

See our response to your first comment.

l.184 "data pre-analyses were performed in MS excel" what are pre-analyses?

The section (lines 225–246) has been revised to clarify this point.

l.185 : put the sentence about R version (Rstudio doesn't matter) at the end of the paragraph.

The sentence has been moved at the end of the section (line 249), as suggested.

l. 200 (line 243–248) : Which package/ function are used for the deviation tests?

The tests were performed in Excel, using the T.DIST.RT() function. We have clarified that in the revised manuscript at line 244.

**c) Results:**

l. 243-245 (lines 294–300): I don't understand the explanation about reduced soil mass, please rephrase.

Thank you for this comment. The sentence refers to the reduced soil mass sampled on day 80, as detailed in the Materials and Methods (line 182). This explanation was already clarified in previous revisions of the manuscript (see lines 294–300).

l. 290: extra point in "…60% of the variance. Separated.."

Thank you for pointing this out. The sentence has been revised during previous rounds of review, and we believe that any typographical issue has been corrected in the current version.

l. 290-300 : "The first dimension, accounting for 60 % of the variance, separated the molecules in two groups: (1) negative values corresponded to substances such as mefentrifluconazole and tebuconazole, which have high soil sorption, high lipophilicity, low water solubility and/or long soil persistence; and (2) positive values corresponded to substances such as clopyralid or pyroxsulam, which have low soil sorption, low lipophilicity, high water solubility and/or short soil persistence. "**1.** It is not entirely clear which data were included in the PCA analysis. Did the authors use only the percentage of the initial pesticide concentration at each sampling date, or were physicochemical properties and sampling compartments also incorporated? In line 302, the statement *"reflecting a shift towards a dominance of molecules with higher soil sorption,*

*bioconcentration or persistence"* is ambiguous, as it is unclear which part of this interpretation is directly supported by the PCA results and which derives from the known properties of the molecules.

> Thank you for this comment. The PCA was performed exclusively on the quantified content of each compound in each individual sample. No physicochemical properties of the compounds were included in the analysis, and sampling compartments or sampling dates were not used as input variables. These metadata were only used a posteriori to colour and annotate the score plot and for interpretation. The variables in the PCA correspond solely to the quantified compounds, and the individuals correspond to the individual samples at each sampling date. We have clarified this in the revised manuscript (lines 342–343) as follows:

> Sampling date, compartment and physicochemical properties were not included as input variables but used only for visual grouping in the score plot.

> The interpretation in lines 341–359 (revised during the previous revisions of the manuscript) therefore combines (i) the structure of the loading plot, which shows how individual compounds drive separation along the principal components, and (ii) the known properties of these compounds, as reported in Table S5. To avoid any ambiguity, we have revised the section to explicitly state that these properties are used only to interpret the PCA, not to compute it. We also now refer directly to Table S5 in the manuscript. We have clarified this in the revised manuscript (lines 344–349) as follows:

> Looking the loading plot (Fig. 2, right panel) and the physicochemical properties of the compounds (Table T5 in the Supplement S1), we see that the first dimension of the PCA, accounting for 60 % of the variance, separated the molecules in two groups: (1) negative values corresponded to substances such as mefentrifluconazole and tebuconazole, which have high soil sorption, high lipophilicity, low water solubility and/or long soil persistence; and (2) positive values corresponded to substances such as clopyralid or pyroxsulam, which have low soil sorption, low lipophilicity, high water solubility and/or short soil persistence.

l. 302 please define "post-emergence"

> The term post-emergence is standard terminology in agronomy and pesticide science and is widely used to describe herbicides applied after crop seeds have germinated and emerged above the soil surface. Considering the targeted audience of the manuscript, we believe the term is sufficiently common and does not require additional definition. We therefore did not modify the manuscript in response to this comment.

> Please note that the section you are referring to is not part of the main text but was shifted in the Supplements during the first round of review.

Fig. 2: I think that it can be useful to have subpanels for left and right. Also, I don't understand what is the right plot.

> Thank you for this helpful comment. The left and right panels correspond to the standard *score plot* and *loading plot* of a PCA, respectively. The score plot displays the observations

(samples) in the principal component space, whereas the loading plot shows how the variables (quantified compounds) contribute to these principal components. To clarify this distinction, we have revised the figure caption as follows:

**Figure 2.** Principal component analysis (PCA) of all quantified samples: we observe that the relative profile of compounds in soil and soil solution samples changed over time. **Left:** score plot of the samples, illustrating their distribution along the first two principal components based on their compound profile. **Right:** loading plot of the quantified compounds, indicating how each contributes to the separation of samples along the first two principal components. The three molecules in bold in the right panel were selected for the individual analysis detailed in Supplements S3 and S4.

l. 320 -340: The interpretations and proposed mechanisms, particularly those related to water fluxes and the effects of crop density, are insufficiently supported by evidence.

Thank you for this comment. This section was already revised during the first round of review; we believe that the concern has been addressed in the revised text. We also invite you to refer to our response to your first comment.

l. 482 (lines 523–527) : "Our main hypothesis highlighted the role of microorganisms in pesticide biodegradation, but we were unable to directly monitor microbial activity ". In my opinion, this sentence-and possibly the introduction as well - should be reformulated to clearly emphasize the main hypothesis and how the study was designed to address it. The statement *"our study was not designed to test our hypothesis"* seems inappropriate, as it undermines the scientific rationale and clarity of the research objective

Thank you for highlighting this important point. We have revised both the introduction (see our response to your first comment) and this section to clarify this. Lines 523–527 now read:

Although our interpretation of pesticide behaviour draws on the widely acknowledged role of rhizosphere-mediated microbial processes in pesticide biodegradation, we were unable to directly monitor microbial activity. Further research integrating both pesticide quantification and microbial activity measurements would provide valuable mechanistic understanding of the processes driving residue dynamics under cover crops.